# The Accumulation of Score Estimation Error in Diffusion Models

**Baoxiang He** [1]   **Valentio Iverson** [2]   **Shuai Li** [1]   **Cheng Chen** [3]   **Bo Jiang** [1]

## Abstract

Diffusion models are widely used for high-quality generation, but their performance is sensitive to the accuracy of the estimated score. We first derive a stepwise Wasserstein error bound in a Gaussian-mixture setting, where the score admits a closed-form structure, and the score Hessian can be controlled explicitly, leading to sharp Wasserstein estimates. We then extend the analysis to general data distributions, which yields a more general but typically looser upper bound. This general bound can be sharpened under mild regularity: when the initial distribution has a globally Lipschitz score, the curvature contribution at small times is uniformly bounded, avoiding the worst-case blow-up. The results hold for both variance-preserving (VP) and variance-exploding (VE) diffusions, and apply to both the reverse-time SDE and the associated probability-flow ODE.

## 1. Introduction

Diffusion models, also known as score-based generative models (Song et al., 2021b), have become a leading paradigm for generative modeling, achieving state-of-the-art results in image synthesis (Rombach et al., 2022; Ramesh et al., 2022; Huang et al., 2025) and video generation (Bar-Tal et al., 2024; Blattmann et al., 2023). Diffusion models consist of two coupled processes: a forward process, which gradually perturbs data by adding noise, and a reverse process, which reconstructs data from Gaussian noise. The reverse dynamics require the score function, i.e., the gradient of the log-density of the perturbed distribution, denoted by $\nabla \log p_t(x)$ where $p_t$ is the marginal law of the forward process at time $t$.

Since the true score is intractable, it is approximated by training neural networks (Salimans & Ho; Song & Ermon, 2019; Ho et al., 2020), and the learned score is then employed to simulate the reverse process through discretized SDE or ODE solvers (see Section 2 for details).

There are two main sources of error in the reverse process: the discretization error, arising from numerical approximation of the dynamics, and the score estimation error, arising from approximating the true score with a learned network. Extensive prior work has focused on analyzing discretization error, i.e., the error introduced by numerically discretizing the reverse process (De Bortoli, 2022; Chen et al., 2023; Benton et al., 2024; Li & Cai, 2024; Li et al., 2025). In these analyses, the learned score $s_\theta(x, t)$ is typically assumed to approximate the true score $\nabla \log p_t(x)$ with a uniform $L^2$ error bounded by $\epsilon_0^2$.

However, existing analyses of score estimation error are rather coarse: its effect on the final distribution is usually bounded by terms of order $T\epsilon_0^2$ (Chen et al., 2023; Benton et al., 2024), which obscure the role of step-size allocation and fail to capture which regions of the time horizon contribute most critically. Prior work has demonstrated that the choice of step sizes has a significant impact on the quality of generated samples (Karras et al., 2022; Lu et al., 2022; Sabour et al., 2024). In particular, although the estimation error at each step may be small, it propagates across the entire reverse trajectory and can substantially degrade sample quality. This issue is especially pronounced in regions of low signal-to-noise ratio (SNR), where empirical evidence shows that score approximation errors are relatively large (Nichol & Dhariwal, 2021; Wu et al., 2024).

Understanding how such errors accumulate under different discretization schemes is therefore essential for explaining the sensitivity of diffusion models to noise schedules and for developing more robust samplers. Motivated by this gap, our work develops a non-asymptotic analysis of score error propagation, yielding theoretical insights that explain observed schedule sensitivity and clarify the roles of discretization strategies and sampling formulations.

Our main contributions are summarized as follows:

- We start from a unified gain recursion based on the pathwise Hessian average, and use it to control Wasser-

[1]John Hopcroft Center for Computer Science, Shanghai Jiao Tong University, Shanghai, China [2]Cheriton School of Computer Science, University of Waterloo, Waterloo, Canada [3]East China Normal University, Shanghai, China. Correspondence to: Bo Jiang <bjiang@sjtu.edu.cn>.

*Proceedings of the 43$^{rd}$ International Conference on Machine Learning*, Seoul, South Korea. PMLR 306, 2026. Copyright 2026 by the author(s).

stein error accumulation along the reverse dynamics.

- In the Gaussian case, the Hessian is available in closed form ($H_k = -\Sigma_{\tau_k}^{-1}$), leading to explicit and interpretable $W_2$ bounds. For Gaussian mixtures, we replace the intractable pathwise Hessians by computable surrogate Hessians (mixture-average or dominant-component), and obtain corresponding Wasserstein bounds.

- For general $p_0$, we give a distribution-agnostic baseline bound via the forward smoothing scale, and refine it under a Lipschitz-score assumption; the same results cover VP/VE and extend to probability-flow ODE discretizations.

## 2. Preliminaries

In this section we provide background on diffusion models, including the forward and reverse processes, score estimation, and sampling methods.

**Forward Process**  The forward process gradually perturbs a clean data point $x_0 \sim p_0$, where $p_0$ is a distribution on $\mathbb{R}^d$. Its evolution is described by the stochastic differential equation

$$dX_t = \beta(t)X_t \, dt + \alpha(t) \, dW_t, \qquad (1)$$

where $(W_t)_{t \geq 0}$ is a standard Brownian motion in $\mathbb{R}^d$, and we denote by $p_t$ the law of $X_t$ for each $t \in [0, T]$.

**Reverse Process**  The reverse process reconstructs data by inverting the forward dynamics. It is initialized from $Y_0 \sim q_0$, where $q_0 = p_T$ is the terminal law of the forward process, and evolves back to a distribution $q_T$ close to the data distribution $p_0$. The reverse-time SDE is given by Anderson (1982); Song et al. (2021b):

$$dY_t = \big(\beta(t)Y_t - \alpha(t)^2 \nabla \log p_t(Y_t)\big)dt + \alpha(t)d\overline{W}_t, \quad (2)$$

where $(\overline{W}_t)_{t \geq 0}$ is a time-reversed Brownian motion, and $\nabla \log p_t(x)$ denotes the score function of $p_t$. By construction, the forward and reverse processes are coupled through their marginals:

$$X_t \sim p_t \qquad \text{and} \qquad Y_t \sim q_t \text{ with } q_t = p_{T-t}.$$

In particular, the forward terminal distribution $p_T$ serves as the initialization $q_0$ for the reverse dynamics, and the reverse terminal distribution $q_T$ recovers the data distribution $p_0$.

**Score Estimation**  In practice, the true score $\nabla \log p_t(x)$ is inaccessible since the marginal distribution $p_t$ is unknown. To address this, one trains a time-dependent neural network $s_\theta(x, t)$ using *denoising score matching* (DSM) (Vincent, 2011; Song & Ermon, 2019). The DSM objective is

$$\min_\theta \; \mathbb{E}_{t,x_0,x_t}\Big[\big\|s_\theta(x_t, t) - \nabla \log p_{t|0}(x_t \mid x_0)\big\|_2^2\Big].$$

Here $t \sim U(0, T)$, $x_0 \sim p_0$, and $x_t \sim p_{t|0}(\cdot \mid x_0)$. Consequently, the learned network $s_\theta$ serves as an approximation to the true score function and can be used as a surrogate for $\nabla \log p_t$ in the reverse-time dynamics.

**Sampling Methods**  Once the score network is trained, new samples are generated by simulating the reverse-time dynamics. This requires discretizing the reverse SDE or ODE. The specific form of the reverse process depends on the choice of the coefficients $(\beta(t), \alpha(t))$ in the forward SDE (1).

Two standard formulations are widely used: the **variance-preserving (VP)** diffusion, with $\beta(t) = -1$ and $\alpha(t) = \sqrt{2}$ (Chen et al., 2023), and the **variance-exploding (VE)** diffusion, with $\beta(t) = 0$ and $\alpha(t) = \sqrt{2}$ (Song et al., 2021b). For these two cases, the forward marginals admit closed-form conditionals:

$$p(x_t \mid x_0) = \begin{cases} \mathcal{N}\big(e^{-t}x_0, \; (1 - e^{-2t})I_d\big), & \text{VP}, \\ \mathcal{N}\big(x_0, \; 2tI_d\big), & \text{VE}. \end{cases}$$

We next introduce the time discretization used for simulating the reverse dynamics. Let $\{h_j\}_{j=0}^{K-1}$ with $h_j > 0$ denote a partition of $[0, T]$ into $K$ steps, and define the forward grid

$$t_k = \sum_{j=0}^{k-1} h_j, \qquad T = \sum_{j=0}^{K-1} h_j.$$

The reverse-time grid is simply the forward grid read backwards:

$$\tau_k = t_{K-1-k}, \qquad h_k^{\leftarrow} = h_{K-1-k}.$$

We use the exponential integrator (EI) sampler (Zhang & Chen, 2023) to illustrate the scheme. The analysis, however, is not specific to EI: the same proof strategy applies to a broad class of first-order samplers (Ho et al., 2020; Song et al., 2021a), with only minor modifications to the coefficients of the update equations.

For the VP-SDE, the reverse update is

$$y_{k+1} = e^{h_k^{\leftarrow}} y_k + 2\big(e^{h_k^{\leftarrow}} - 1\big)\nabla \log p_{\tau_k}(y_k) + \sqrt{e^{2h_k^{\leftarrow}} - 1}\, z_k, \tag{3}$$

with initialization $y_0 \sim \mathcal{N}(0, I_d)$ and Gaussian noise $z_k \sim \mathcal{N}(0, I_d)$.

For the VE-SDE, the corresponding reverse update is

$$y_{k+1} = y_k + 2h_k^{\leftarrow}\nabla \log p_{\tau_k}(y_k) + \sqrt{2h_k^{\leftarrow}}\, z_k, \tag{4}$$

with initialization $y_0 \sim \mathcal{N}(0, 2TI_d)$ and $z_k \sim \mathcal{N}(0, I_d)$.

Higher-order samplers can be analyzed in a similar way, but this typically requires controlling higher-order derivatives of the score. We leave this direction to future work.

# 3. Main Results

We first introduce a standard time-resolved regularity assumption on the score approximation error.

**Assumption 3.1** (Score approximation error with time-resolved regularity). Let $e(x, t) := s_\theta(x, t) - \nabla \log p_t(x)$ denote the score approximation error. There exist nonnegative functions $t \mapsto L_t$ and $t \mapsto \varepsilon_t$ on $[0, T]$ such that, for every $t \in [0, T]$,

(i) For each $t \in [0, T]$, the map $x \mapsto e(x, t)$ is $L_t$-Lipschitz:

$$\|e(x, t) - e(y, t)\| \leq L_t \|x - y\|, \qquad \forall x, y \in \mathbb{R}^d.$$

Moreover, the profile $t \mapsto L_t$ is non-increasing on $[0, T]$ (i.e., $L_{t_1} \geq L_{t_2}$ for all $0 \leq t_1 \leq t_2 \leq T$).

(ii) For each $t \in [0, T]$, the error has bounded second moment under $p_t$:

$$\mathbb{E}_{x \sim p_t}\big[\|e(x, t)\|^2\big] \leq \varepsilon_t^2.$$

Moreover, the profile $t \mapsto \varepsilon_t$ is non-increasing on $[0, T]$ (i.e., $\varepsilon_{t_1} \geq \varepsilon_{t_2}$ for all $0 \leq t_1 \leq t_2 \leq T$).

In particular, the slice-wise $L^2$ control in (ii) is standard in diffusion-model analyses and underlies many sampling error bounds (Chen et al., 2023; Benton et al., 2024). Moreover, when $p_0$ is a Gaussian mixture, the ground-truth score $\nabla \log p_t$ is globally Lipschitz for every $t > 0$; thus condition (i) holds whenever $s_\theta(\cdot, t)$ is Lipschitz; see Lemma B.3 in Appendix B.

The monotone profiles of $L_t$ and $\varepsilon_t$ formalize a common empirical trend across noise levels. As $t$ increases, the marginal $p_t$ is more heavily smoothed (and typically closer to a Gaussian), so the score field $\nabla \log p_t$ varies more gently and is easier to approximate. Consequently, the score network tends to incur smaller errors, leading to smaller $\varepsilon_t$ and Lipschitz constants $L_t$ at the noise end. Conversely, errors often concentrate near the data end (small $t$), where the landscape is sharper and approximation is more challenging (Nichol & Dhariwal, 2021).

With Assumption 3.1, the perturbed reverse updates for VP/VE take the form

$$y_{k+1} = e^{h_k^\leftarrow} y_k + 2\big(e^{h_k^\leftarrow} - 1\big)\Big(\nabla \log p_{\tau_k}(y_k) + e(y_k, \tau_k)\Big)$$
$$+ \sqrt{e^{2h_k^\leftarrow} - 1}\, z_k. \tag{5}$$

with initialization $y_0 \sim \mathcal{N}(0, I_d)$. The corresponding baseline trajectory $\{y_k^{(0)}\}$ is obtained by removing the error term $e(y_k, \tau_k)$. For the VE-SDE, the update is

$$y_{k+1} = y_k + 2h_k^\leftarrow \Big(\nabla \log p_{\tau_k}(y_k) + e(\tau_k, y_k)\Big) + \sqrt{2h_k^\leftarrow}\, z_k, \tag{6}$$

with initialization $y_0 \sim \mathcal{N}(0, 2TI_d)$ for horizon $T > 0$. Again, the baseline sequence $\{y_k^{(0)}\}$ is defined analogously by removing $e(y_k, \tau_k)$. We denote by $p(y_K) := \mathcal{L}(y_K)$ and $p(y_K^{(0)}) := \mathcal{L}(y_K^{(0)})$ the terminal laws of the perturbed and baseline updates, respectively.

We first introduce the linearized difference dynamics between the approximate sampler (5)–(6) and the oracle sampler (3)–(4) under synchronous coupling.

To express the local linearization of the score field along the coupled paths, we start by defining a pathwise Hessian average.

**Definition 3.2** (Pathwise Hessian average). For each $k = 0, \ldots, K - 1$, define the pathwise Hessian average along the line segment connecting $y_k^{(0)}$ and $y_k$ by

$$H_k := \int_0^1 \nabla^2 \log p_{\tau_k}\Big(y_k^{(0)} + t\big(y_k - y_k^{(0)}\big)\Big)\, dt. \tag{7}$$

Equivalently, the score increment admits the exact linear representation

$$\nabla \log p_{\tau_k}(y_k) - \nabla \log p_{\tau_k}(y_k^{(0)}) = H_k\, (y_k - y_k^{(0)}).$$

The matrix $H_k$ captures the local curvature of $\log p_{\tau_k}$ along the realized segment. To convert this curvature information into a tractable bound on the evolution of $\|\Delta_k\|$, we introduce stepwise growth factors and their cumulative products.

**Definition 3.3** (Growth factors). For each reverse step $j = 0, \ldots, K - 1$, define the scalar growth factor

$$\lambda_j(H_j) := \|\alpha_j I_d + \beta_j H_j\|_{\text{op}} + \beta_j L_{\tau_j}. \tag{8}$$

Given $H := \{H_j\}_{j=0}^{K-1}$, define the cumulative scalar gain to the terminal step by

$$g_i(H) := \left(\prod_{j=i+1}^{K-1} \lambda_j(H_j)\right)\beta_i, \qquad i = 0, \ldots, K-1, \tag{9}$$

with the convention that the empty product equals 1.

Here $(\alpha_j, \beta_j)$ are sampler-dependent coefficients: $(\alpha_j, \beta_j) = (e^{h_j^\leftarrow}, 2(e^{h_j^\leftarrow} - 1))$ for VP-SDE and $(\alpha_j, \beta_j) = (1, 2h_j^\leftarrow)$ for VE-SDE. When the dependence on the curvature collection $H = \{H_j\}_{j=0}^{K-1}$ is clear from context, we write $\lambda_j$ and $g_i$ as shorthand for $\lambda_j(H_j)$ and $g_i(H)$, respectively.

With these definitions in place, the coupled difference process admits a simple one-step bound.

**Lemma 3.4** (Error recursion)*. Let $\{y_k\}_{k=0}^K$ and $\{y_k^{(0)}\}_{k=0}^K$ be synchronously coupled trajectories of the approximate and oracle samplers, respectively, and define $\Delta_k := y_k - y_k^{(0)}$. Then for each $k = 1, \ldots, K$,*

$$\|\Delta_k\| \le \lambda_{k-1}\,\|\Delta_{k-1}\| + \beta_{k-1}\,\Big\|e(\tau_{k-1}, y_{k-1}^{(0)})\Big\|, \quad (10)$$

*where $\lambda_{k-1}$ is defined in Definition 3.3.*

Lemma 3.4 separates the evolution into two effects: a multiplicative amplification term governed by curvature (via $H_{k-1}$) and an additive injection term due to the error term $e(\tau_{k-1}, y_{k-1}^{(0)})$ of the local score error. The main obstacle is that the exact pathwise Hessian $H_k$ in (7) depends on the unknown marginal $p_{\tau_k}$ and the realized trajectories, and is therefore not directly tractable.

In the remainder of the paper, we bound $H_k$ by exploiting the structure of the forward marginals. The different distributional assumptions and their corresponding Hessian control mechanisms are summarized in Table 1. In the Gaussian case, $H_k$ has a closed-form expression, which yields explicit gains and a sharp Wasserstein bound (Section 3.1). For Gaussian mixtures, we introduce tractable surrogate Hessians and show they provide reliable curvature control across different separation regimes (Section 3.2). For general initial laws, we revert to operator-norm curvature envelopes, leading to conservative but fully distribution-agnostic Wasserstein estimates (Section 3.3). Under an additional Lipschitz-score condition on $p_0$, we further refine the small-time Hessian control in expectation (Corollary 3.14). We also extend the framework to probability-flow ODE discretizations (Section 3.4).

*Table 1.* Summary of Hessian control mechanisms under various assumptions.

| Assumption | Hessian Control |
|---|---|
| Gaussian $p_0$ | Exact pathwise Hessian via $\Sigma_t^{-1}$ (Thm. 3.5) |
| Gaussian mixture | Surrogate Hessian + correction term (Thm. 3.7) |
| Separation regimes | Localized, regime-adapted surrogate (Thm. 3.11) |
| General $p_0$ | Pathwise Hessian surrogate bound (Thm. 3.12) |
| Lipschitz score | Hessian expectation at small-time (Cor. 3.14) |

### 3.1. Gaussian Distribution

Before presenting the general results, we first highlight the mechanism in the simple case where the initial distribution for the forward process is Gaussian distribution on $\mathbb{R}^d$:

$$p_0 = \mathcal{N}(\mu_0, \Sigma_0), \qquad \mu_0 \in \mathbb{R}^d,\ \Sigma_0 \in \mathbb{R}^{d \times d},\ \Sigma_0 \succeq 0.$$

Under the forward VP/VE processes the distribution remains Gaussian, and the score admits the closed form

$$\nabla_x \log p_t(x) = -\Sigma_t^{-1}(x - \mu_t), \quad (11)$$

where $(\mu_t, \Sigma_t)$ correspond to either VP or VE dynamics, as given in (12).

$$\mu^{\mathrm{VP}}(t) = e^{-t}\mu_0, \qquad \Sigma_i^{\mathrm{VP}}(t) = e^{-2t}\Sigma_0 + (1 - e^{-2t})I_d,$$
$$\mu^{\mathrm{VE}}(t) = \mu_0, \qquad \Sigma_i^{\mathrm{VE}}(t) = \Sigma_0 + 2tI_d. \quad (12)$$

We note that the Gaussian score (11) is linear in $x$, and therefore the pathwise Hessian averages in (7) reduce to the exact Hessians

$$H_k = -\Sigma_{\tau_k}^{-1}.$$

Then we obtain Theorem 3.5, which provides an upper bound on the Wasserstein distance between the terminal laws of the perturbed and baseline dynamics. The proof is deferred to Appendix A.

**Theorem 3.5** (Wasserstein Error Bound)*. Under Assumption 3.1 and the Gaussian score representation (11), the terminal laws $p(y_K)$ and $p(y_K^{(0)})$ satisfy:*

$$W_2^2\big(p(y_K),\, p(y_K^{(0)})\big) \le K \sum_{i=0}^{K-1} g_i^2 \varepsilon_{\tau_i}^2. \quad (13)$$

*Remark* 3.6. The bound (13) shows that the terminal Wasserstein error is governed by two ingredients: the amplification factors $g_i$ and the local error magnitudes $\varepsilon_{\tau_i}$. The operators $g_i$ depend on the discretization schedule, the curvature matrices $H_j = -\Sigma_{\tau_j}^{-1}$, and the Lipschitz constants $L_{\tau_j}$, and therefore describe how perturbations propagate along the reverse dynamics. As a result, both the geometry of $p_t$ (via $H_j$) and the regularity of the learned score (via $L_{\tau_j}$) shape how local errors accumulate over time.

To further illustrate Theorem 3.5, consider the isotropic Gaussian case where $p_0 = \mathcal{N}(\mu_0, \sigma_0^2 I_d)$. Under the small step-size regime ($h_j^{\leftarrow} \ll 1$), the bound in (13) simplifies to:

$$W_2^2\big(p(y_K),\, p(y_K^{(0)})\big) \le K \sum_{i=0}^{K-1} \beta_i^2\, \Gamma_i^2\, \epsilon_{\tau_i}^2, \quad (14)$$

where

$$\Gamma_i := \exp\!\left(\sum_{j=i+1}^{K-1} h_j^{\leftarrow} c_{\tau_j}\right).$$

Here $c_\tau$ is determined by the VP/VE processes as

$$c_\tau = \begin{cases} 1 - 2\big(1 - (1 - \sigma_0^2)e^{-2\tau}\big)^{-1} + 2L_\tau, & \text{VP-SDE}, \\ -2\big(\sigma_0^2 + 2\tau\big)^{-1} + 2L_\tau, & \text{VE-SDE}. \end{cases}$$

The bound in (14) separates the terminal error into a sum of local score errors, each weighted by an amplification

factor $\Gamma_i$ that depends on the remaining reverse trajectory. In the Gaussian case, this weighting is governed by the time-varying curvature through $\Sigma_t^{-1}$, together with the time profiles of the approximation error $(\varepsilon_t)$ and the Lipschitz constants $(L_t)$.

With Assumption 3.1, both $L_t$ and $\varepsilon_t$ are near the data end($t=0$). This makes the late reverse steps the most sensitive regime: even if the remaining product defining $\Gamma_i$ is shorter, the injected errors themselves are typically larger and more state-dependent. This motivates allocating finer steps near $t = 0$ to control these large local errors, while using coarser steps in the high-noise regime where $\varepsilon_t$ and $L_t$ are smaller.

As a concrete illustration, consider the VE case in the isotropic setting. When $L_\tau$ is moderate, the curvature term dominates and

$$c_\tau = -2(\sigma_0^2 + 2\tau)^{-1} + 2L_\tau \approx -\frac{1}{\tau} < 0.$$

Consequently,

$$\Gamma_i = \exp\left(\sum_{j=i+1}^{K-1} h_j^\leftarrow c_{\tau_j}\right) \leq 1,$$

and since the exponent is a sum of negative terms, $\Gamma_i$ is typically *smaller* for indices $i$ in the noise end (large $\tau_i$) and closer to one near the data end (small $\tau_i$).

This comparison indicates that errors injected near the data end are therefore more likely to accumulate, motivating smaller steps in the low-noise region.

### 3.2. Gaussian Mixtures

The Gaussian case in Theorem 3.5 provides a clean expression where the linear structure of the score (11) leads directly to the path Hessian average. This toy example illustrates the central mechanism by which score perturbations propagate through the dynamics.

We now extend the analysis to the more general and practically relevant case where the initial distribution is a mixture of Gaussians. In this setting, the score is no longer linear in $x$, yet the mixture structure still enables meaningful control of the induced Wasserstein error. Specifically, let the initial distribution be a Gaussian mixture on $\mathbb{R}^d$:

$$p_0(x) = \sum_{m=1}^M \pi_m \mathcal{N}\big(x; \mu_m(0), \Sigma_m(0)\big), \qquad (15)$$

where $\pi_m \geq 0$, $\sum_{m=1}^M \pi_m = 1$, $\mu_m(0) \in \mathbb{R}^d$, and $\Sigma_m(0) \in \mathbb{R}^{d \times d}$ are symmetric positive semidefinite. Denote by $p_t$ the forward law at time $t$. As in the Gaussian case, under VP/VE diffusions each mixture component evolves

according to (12), i.e.,

$$p_t(x) = \sum_{m=1}^M \pi_m \mathcal{N}\big(x; \mu_m(t), \Sigma_m(t)\big).$$

For a Gaussian mixture $p_{\tau_k}(x) = \sum_{m=1}^M \pi_m \mathcal{N}\big(x; \mu_m(\tau_k), \Sigma_m(\tau_k)\big)$, it admits the decomposition

$$\nabla^2 \log p_{\tau_k}(x) = -\sum_{m=1}^M \gamma_m(x; \tau_k) \, \Sigma_m(\tau_k)^{-1} \\ + \mathrm{Cov}_{m \sim \gamma(\cdot \,|\, x; \tau_k)}\big[v_m(x; \tau_k)\big], \qquad (16)$$

where

$$\gamma_m(x; \tau_k) := \frac{\pi_m \mathcal{N}\big(x; \mu_m(\tau_k), \Sigma_m(\tau_k)\big)}{\sum_{j=1}^M \pi_j \mathcal{N}\big(x; \mu_j(\tau_k), \Sigma_j(\tau_k)\big)},$$

and

$$v_m(x; \tau_k) := \Sigma_m(\tau_k)^{-1}\big(\mu_m(\tau_k) - x\big).$$

We then apply Lemma 3.4 to obtain Wasserstein error bounds involving the pathwise Hessians $H_k$ defined in (7).

In practice, however, the exact pathwise Hessians $H_k$ in (7) are not available. We therefore introduce two computable surrogates: an *average surrogate*, obtained by weighting component Hessians by their mixture weights, and a *dominant-component surrogate*, obtained by taking the Hessian of the most likely component at $y_k$:

$$\bar{H}_k^{\mathrm{ave}} := -\sum_{m=1}^M \pi_m \, \Sigma_m(\tau_k)^{-1}, \\ \bar{H}_k^{\mathrm{dom}} := -\Sigma_{i^\star(y_k)}(\tau_k)^{-1}, \qquad (17)$$

where $i^\star(y_k) := \arg\max_{m \in [M]} \gamma_m(y_k; \tau_k)$.

We next show that replacing the exact $H_k$ with either surrogate still yields a valid Wasserstein error bound, with the guarantee depending on the tighter of the two choices.

**Theorem 3.7** (GM bound with surrogate Hessians). *Let $g_i(\cdot)$ be defined in Definition 3.3. Under Assumption 3.1, and assuming the forward initial law is the Gaussian mixture in (15), the terminal laws of the perturbed and baseline updates satisfy*

$$W_2^2\big(p_K, p_K^{(0)}\big) \leq \min_{r \in \{\mathrm{ave,\,dom}\}} \sum_{i=0}^{K-1} g_i^2\big(\bar{H}_i^r\big) \epsilon_{\tau_i}^2 + \widehat{\Delta},$$

$$\widehat{\Delta} \leq C\left(\sum_{i=0}^{K-1} \beta_i \sum_{j=i+1}^{K-1} \beta_j \,(d+2)\,\Lambda_j\right) \mathcal{S}_0, \qquad (18)$$

*where*

$$\mathcal{S}_0 := \max_{0 \leq i \leq K-1} \epsilon_{\tau_i}, \qquad \Lambda_j := \max_{m \in [K]} \|\Sigma_m(\tau_j)^{-1}\|_{\mathrm{op}}.$$

Here $(\alpha_j, \beta_j)$ in $g_i(\cdot)$ are those of the chosen VP/VE sampler (cf. (9)), and $C > 0$ is an absolute constant independent of $K$ and $d$.

The proof is deferred to Appendix A.

*Remark* 3.8. Theorem 3.7 decomposes the terminal error into a leading accumulation term and a surrogate-Hessian mismatch term $\widehat{\Delta}$. The leading term shows that local score errors $\epsilon_{\tau_i}$ are weighted by the schedule-dependent gains $g_i(\bar{H}_i^r)$, which depend on the curvature of the Gaussian-mixture marginals through the surrogate Hessians. The correction term $\widehat{\Delta}$ measures the price of replacing the exact pathwise Hessian by the average or dominant-component surrogate.

This decomposition gives a concrete explanation for data-end refinement. Near the data end, the marginal is less smoothed, the score field can vary more rapidly, and the score-estimation error is typically larger. Hence errors introduced at small $\tau_i$ can contribute more strongly to the final Wasserstein error. Near the noise end, the distribution is more regular and the surrogate Hessians provide more stable curvature control, so larger steps are less harmful. Consequently, schedules that allocate smaller steps near $t = 0$ are favored by the bound in (18).

**The choices of $\bar{H}_k$.** We now turn to the choice of the surrogate Hessian $\bar{H}_k$. The appearance of the minimum in (18) reflects that, depending on the geometry of the Gaussian mixture, either the mixture-weighted surrogate $\bar{H}^{\mathrm{ave}}$ or the dominant-component surrogate $\bar{H}^{\mathrm{dom}}$ may yield a tighter control of the error.

We distinguish two regimes that guide the choice of the surrogate $\bar{H}_k$:

**Definition 3.9** (Small separation)**.** Define the mean separation

$$\delta_\mu(t) := \max_{m \neq n} \left\| \Sigma_m(t)^{-1/2} \big( \mu_m(t) - \mu_n(t) \big) \right\|,$$

and the covariance separation

$$\delta_\Sigma(t) := \max_{m,n} \left\| \Sigma_m(t)^{1/2} \Sigma_n(t)^{-1} \Sigma_m(t)^{1/2} - I_d \right\|_{\mathrm{op}}.$$

Let $\delta(t) := \max\{\delta_\mu(t), \delta_\Sigma(t)\}$. We say the mixture is in the *small separation regime* at time $t$ if $\delta(t) \ll 1$.

**Definition 3.10** (Large separation)**.** For $x \in \mathbb{R}^d$, define the logits

$$\ell_i(x) = \log \pi_i - \tfrac{1}{2} \log \det(2\pi \Sigma_i(t))$$
$$- \tfrac{1}{2}(x - \mu_i(t))^\top \Sigma_i(t)^{-1}(x - \mu_i(t)).$$

Let $i^*(x) = \arg\max_i \ell_i(x)$ and the logit margin

$$\kappa_t(x) := \min_{j \neq i^*(x)} \big( \ell_{i^*}(x) - \ell_j(x) \big).$$

We say the mixture is in the *large separation regime* along a path $\{x_t\}$ if $\kappa_{\tau_k}(x_t) \geq \underline{\kappa} \gg 1$ for all $t \in [0,1]$.

We give Theorem 3.11 to refine Theorem 3.7 by adapting the surrogate Hessian according to the separation regime of the mixture. The proof of Theorem 3.11 is deffered to Appendix A.

**Theorem 3.11.** *Under the same setting as Theorem 3.7, let $p_{\tau_k}$ denote the marginal distribution of the forward process at time $\tau_k$. Define*

$$K_S := \max\{ k : p_{\tau_k} \text{ lies in the small separation regime} \},$$

$$K_L := \min\{ k : p_{\tau_k} \text{ lies in the large separation regime} \}.$$

*Then, with regime-adapted choices of $\bar{H}_k$, the error term $\widehat{\Delta}$ in* (18) *satisfies*

$$
\begin{aligned}
\widehat{\Delta} \leq{} & \sum_{k=0}^{K_S} O\big(\delta(\tau_k)\big) + \sum_{k=K_L}^{K} O\big(e^{-\underline{\kappa}}\big) \\
& + \sum_{k=K_S+1}^{K_L-1} \left( \beta_k \sum_{j=k+1}^{K-1} \beta_j (d+2) \Lambda_j \right) \mathcal{S}_0.
\end{aligned}
\tag{19}
$$

This result highlights that the surrogate choice for $\bar{H}_k$ can be made adaptively: when the mixture is in the small separation regime, averaging across mixture components provides a reliable surrogate; when it is in the large separation regime, the dominant-component surrogate more closely matches the true Hessian. Both cases yield substantially sharper error control than the crude uniform bound. In particular, the error contributions scale as $O(\delta(\tau_i))$ in small-separation regions and decay exponentially in $\underline{\kappa}$ in large-separation regions. Only in intermediate cases where the mixture is neither clearly separated nor overlapping, do we still have the coarse $(d+2)\Lambda_j$ bound.

Moreover, this refinement connects directly to the properties of the initial distribution $p_0(x)$. If $p_0(x)$ is in the small separation regime, then $\widehat{\Delta}$ can be controlled at order $O(\delta)$. If $p_0(x)$ is instead in the large separation regime, and the error perturbations $e_\tau$ are concentrated only near the data end (i.e., at small diffusion times), then $\widehat{\Delta}$ can be controlled at order $O(e^{-\underline{\kappa}})$. Consequently, in these settings the leading terms in (18) provide an accurate reflection of the Wasserstein discrepancy, with $\widehat{\Delta}$ reduced to a negligible correction.

### 3.3. General Distributions

The Gaussian and Gaussian-mixture cases show that structural assumptions on the data distribution can yield sharp and interpretable error bounds. We now state a more general result that applies to arbitrary data distributions without requiring such assumptions.

**Theorem 3.12.** *Consider the VP/VE reverse recursions* (5)–(6) *under synchronous coupling. Let* $p_K = \mathcal{L}(y_K)$ *and* $p_K^{(0)} = \mathcal{L}(y_K^{(0)})$ *denote the terminal laws of the perturbed and baseline updates, respectively. Under Assumption 3.1, the terminal Wasserstein deviation satisfies*

$$W_2^2(p_K, p_K^{(0)}) \ \leq \ K \sum_{i=1}^{K-1} g_i^2(H)\, \varepsilon_{\tau_i}^2, \qquad (20)$$

*where* $g_i(H)$ *is defined in* (9), *with* $H$ *taken as*

$$H_j = \begin{cases} \dfrac{d}{\sigma_{\mathrm{VP}}(\tau_j)^2}\, I_d, & \text{VP-SDE}, \\[2mm] \dfrac{d}{\sigma_{\mathrm{VE}}(\tau_j)^2}\, I_d, & \text{VE-SDE}, \end{cases}$$

*and* $\sigma_{\mathrm{VP}}(\tau) = \sqrt{1 - e^{-2\tau}}$, $\sigma_{\mathrm{VE}}(\tau) = \sqrt{2\tau}$ *denote the forward smoothing scales.*

Theorem 3.12 shows that even without structural assumptions, a non-asymptotic Wasserstein bound can be obtained by controlling the curvature of the forward marginals through their smoothing scales. This bound is necessarily conservative: as $\tau \to 0$, the forward variance vanishes and $\sigma_{\mathrm{VP/VE}}(\tau) \to 0$, causing $H$ to blow up. Near the noise end, $\phi_T^{\mathrm{VP}} \approx -1$ and $\phi_T^{\mathrm{VE}} = -1/T < 0$, so amplification is weak and large steps are safe. Near the data end, curvature can be large, and small steps are essential. For these reasons, Theorem 3.12 is stated without structural assumptions: it serves as a worst-case baseline showing that the data-end region is inherently more sensitive to score-estimation errors.

The conservativeness stems from the fact that the curvature is controlled pointwise via the smoothing scale. When $p_0$ enjoys additional regularity—in particular, a globally Lipschitz score and a bounded second moment—the small-time curvature need not blow up, and one can replace the $O(1/\tau)$ worst-case control by a finite constant in expectation. We formalize these conditions as follows.

**Assumption 3.13** (Smooth positive density with Lipschitz score). Let $p_0$ be a density on $\mathbb{R}^d$ and set $\ell_0 := \log p_0$. Assume:

1. $p_0(x) > 0$ for all $x$ and $\ell_0 \in C^2(\mathbb{R}^d)$;

2. $\nabla \ell_0$ is globally $L$-Lipschitz, i.e. $\|\nabla^2 \ell_0(x)\|_{\mathrm{op}} \leq L$ for all $x$;

3. $X_0 \sim p_0$ has finite second moment $\mathbb{E}\|X_0\|^2 \leq M_2 < \infty$.

Under Assumption 3.13, the following result refines the curvature term used in Theorem 3.12 on a small-time interval.

**Corollary 3.14** (Refined Hessian bound in expectation). *Suppose Assumption 3.13 holds. Let*

$$X_t = \mu_t X_0 + \sigma_t Z, \qquad Z \sim \mathcal{N}(0, I_d) \text{ independent of } X_0,$$

*with* $\sigma_t \to 0$ *and* $\mu_t \to 1$ *as* $t \downarrow 0$. *Then there exist* $t_0 > 0$ *and* $m > 0$ *such that* $\mu_t \geq m$ *for all* $t \in (0, t_0]$, *and*

$$\sup_{0 < t \leq t_0} \mathbb{E}\big\|\nabla^2 \log p_t(X_t)\big\|_{\mathrm{op}} \ \leq \ C_0,$$

*where*

$$C_0 := \frac{L + \big(\|\nabla \ell_0(0)\| + L\sqrt{M_2}\big)^2}{m^2} < \infty.$$

*Consequently, in Theorem 3.12, the curvature contribution may be bounded by the piecewise envelope*

$$\overline{H}(\tau) := \begin{cases} C_0, & 0 < \tau \leq t_0, \\[2mm] \dfrac{d}{\sigma_{\mathrm{VP}}(\tau)^2}, & \tau > t_0, \ \text{VP-SDE}, \\[2mm] \dfrac{d}{\sigma_{\mathrm{VE}}(\tau)^2}, & \tau > t_0, \ \text{VE-SDE}, \end{cases}$$

*in the sense that the operator-norm terms involving* $H_j$ *can be replaced by* $\overline{H}(\tau_j)$. *Here* $\sigma_{\mathrm{VP}}(\tau) = \sqrt{1 - e^{-2\tau}}$ *and* $\sigma_{\mathrm{VE}}(\tau) = \sqrt{2\tau}$.

*Remark* 3.15. Theorem 3.12 bounds the Wasserstein error using a curvature envelope that depends only on the smoothing scale and therefore blows up as $\tau \downarrow 0$. Corollary 3.14 shows that under the additional regularity of Assumption 3.13, this small-time blow-up can be avoided in expectation: for $\tau \leq t_0$ the curvature term can be controlled by the finite constant $C_0$. The derivation follows by inserting Lemma B.4 into the proof of Theorem 3.12.

The threshold $t_0$ balances two competing effects. For $\tau \leq t_0$ we use the constant bound $C_0$, which scales like $1/m^2$ with $m := \inf_{t \in (0, t_0]} \mu_t$ (for VP, $\mu_t = e^{-t}$ so $m = e^{-t_0}$ and larger $t_0$ makes $C_0$ larger). For $\tau > t_0$ we revert to the generic bound $1/\sigma_{\mathrm{VP/VE}}(\tau)^2$, which decreases as $\tau$ increases. Thus, increasing $t_0$ reduces the $1/\sigma^2$ term near the switch point but worsens $C_0$; $t_0$ is chosen to trade off these two effects.

### 3.4. Extension to PF-ODE

We now extend the result to the probability-flow ODE (PF-ODE) formulation of diffusion models. Equivalently, the reverse dynamics for (2) can be written as a probability-flow ODE with the same marginals (Song et al., 2021b):

$$dY_t^{\leftarrow} = \Big(\beta(t)Y_t^{\leftarrow} - \tfrac{1}{2}\alpha(t)^2 \nabla_{\boldsymbol{x}} \log p_t(Y_t^{\leftarrow})\Big)\, dt. \quad (21)$$

For concreteness, consider the VP case, whose reverse update reads

$$y_{k+1} = e^{h_k^{\leftarrow}} y_k + \big(e^{h_k^{\leftarrow}} - 1\big)\Big(s_{\tau_k}(y_k) + e(y_k, \tau_k)\Big),$$

with $y_0 \sim \mathcal{N}(0, I_d)$. The baseline trajectory $\{y_k^{(0)}\}$ is obtained by removing $e(y_k, \tau_k)$.

**Corollary 3.16.** *Under the same setting and notation as Theorem 3.12 (in particular $g_i(H)$ as in (9)), the probability-flow ODE discretization satisfies*

$$W_2^2(p_K, p_K^{(0)}) \leq \sum_{i=0}^{K-1} g_i^2(H)\epsilon_{\tau_i}^2, \qquad (22)$$

*where the only change relative to the SDE case lies in the amplification coefficients in $g_i(\cdot)$:*

$$(\alpha_j, \beta_j) = \begin{cases} (e^{h_j^{\leftarrow}}, e^{h_j^{\leftarrow}} - 1), & \text{VP-PF-ODE,} \\ (1, h_j^{\leftarrow}), & \text{VE-PF-ODE.} \end{cases}$$

*Remark* 3.17. This extension shows that our framework applies uniformly to both SDE- and ODE-based samplers. The bias–variance decomposition of the Wasserstein error remains unchanged, and the only difference arises from the amplification coefficients $(\alpha_j, \beta_j)$ encoded in $g_i(H)$.

## 4. Experiments

To empirically illustrate the Wasserstein error bounds derived in Theorem 3.7, we compare the theoretical predicted error accumulation against the actual sampling performance of DDPM (Ho et al., 2020) under various step-size schedules. We construct a controlled environment using a symmetric Gaussian mixture data distribution in $\mathbb{R}^2$, defined as:

$$p_0(x) = \frac{1}{4}\sum_{k=1}^{4} \mathcal{N}(x; \mu_k, \sigma^2 I_2),$$

where the four component means $\{\mu_k\}_{k=1}^4$ are symmetrically distributed in the plane with uniform mixture weights.

For each sampling schedule, we independently train the neural score network five times to approximate the score of the corresponding VP forward process. For each trained model, we plug the empirical score-estimation error into our error-propagation formula (18) to compute the *predicted error accumulation*. For each independently trained score model, we run the reverse-time sampler with the same schedule used in the prediction step and measure the final *sliced $W_2$ distance* between the generated samples and independent samples from the true data distribution. Thus, for each schedule, the reported predicted accumulation and sliced $W_2$ are computed from the same five training runs, and we report their mean and standard deviation.

As reported in Table 2, the average score-estimation $L_2$ error alone does not fully determine the final sampling discrepancy. For example, although the Quadratic schedule achieves the smallest average score error, it produces a larger predicted accumulation and a worse final sliced $W_2$ distance than the Uniform-logSNR schedule. This suggests that the final distributional error is affected not only by the average magnitude of the score error, but also by how such

*Table 2.* Quantitative comparison across different sampling schedules. Score Err. denotes the average score-estimation $L_2$ error, Pred. Acc. denotes the error accumulation predicted by our propagation bound, and Sliced $W_2$ denotes the final sliced Wasserstein distance between generated and true samples. We report the mean and standard deviation over five independent runs. Lower values indicate better performance.

| Schedule | Score Err. | Pred. Acc. | Sliced $W_2$ |
|---|---|---|---|
| Linear | $0.123 \pm 0.032$ | $1.210 \pm 0.301$ | $0.577 \pm 0.127$ |
| Quadratic | $\mathbf{0.113 \pm 0.024}$ | $1.390 \pm 0.502$ | $0.677 \pm 0.082$ |
| Cosine | $0.175 \pm 0.055$ | $2.480 \pm 0.768$ | $0.736 \pm 0.322$ |
| Unif.-logSNR | $0.121 \pm 0.018$ | $\mathbf{1.160 \pm 0.294}$ | $\mathbf{0.545 \pm 0.182}$ |

errors are distributed across time and propagated through the reverse process. In this experiment, the predicted accumulations from our propagation formula are consistent with the observed sliced $W_2$ discrepancies, supporting the qualitative role of the proposed bound in describing how score-estimation errors accumulate during reverse sampling.

To examine this further, Figure 1 illustrates the step-error density predicted by our theory, the integral of which yields the predicted accumulation values reported in Table 2. The plotted density explicitly demonstrates how specific schedules distribute estimation errors over time.

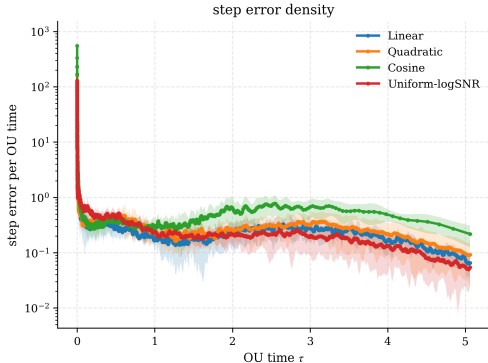

*Figure 1.* Step-wise error density induced by learned score-estimation errors. For each schedule, the empirical score-estimation errors are evaluated through our error-propagation formula to illustrate their local contributions to the final $W_2$ bound.

These results are intended only as a controlled check that the qualitative trends predicted by our bounds can be observed empirically. They should not be interpreted as a claim of schedule optimality, since practical performance can depend on the data distribution, training objective, sampling method, and evaluation metric. The favorable behavior of Uniform-logSNR in this controlled setting is in line with the intuition behind DPM-Solver (Lu et al., 2022), which uses the log-SNR variable $\lambda_t = \log(\alpha_t/\sigma_t)$ to improve numerical integration for diffusion ODEs. This raises the interesting possibility that, beyond its role in numerical discretization, the log-SNR variable may provide a useful time coordinate for understanding both training behavior across

noise levels and the accumulation of score-estimation errors during reverse sampling. A systematic study of this perspective in more realistic settings is left for future work.

## 5. Conclusion

In this work, we analyzed how score estimation errors propagate through the reverse dynamics of diffusion models for both VP and VE processes under reverse SDE and PF-ODE. Starting from the Gaussian case, Theorem 3.5 provided an upper bound on the Wasserstein distance induced by score error, highlighting how discretization steps and the covariance jointly govern error accumulation. For Gaussian mixtures, Theorem 3.7 established a general bound, which can be further tightened under small- or large-separation conditions, thereby adapting to the geometry of the mixture components. Finally, Theorem 3.12 extended the framework to arbitrary data distributions, offering distribution-free but necessarily conservative guarantees. We also give refined bounds under smoothness assumptions on the data distribution in Corollary 3.14. Beyond the theory, we also provide empirical evaluations showing that the resulting bounds track the relative performance of different discretization schedules and are consistent with the observed benefits of data-end–refining step-size choices. Our analysis provides concrete insights into step-size allocation. Near the data end ($t = 0$), where bias is most pronounced, finer discretization is essential to suppress error accumulation, whereas near the noise end ($t = T$) larger steps can be safely used since amplification is weaker.

**Future Work.** This work has focused on how discretization schedules influence the propagation of score-estimation errors during sampling. An important next step is to extend this perspective to the training stage, where the choice of noise schedule also plays a critical role in learning the score function (Hang et al., 2023; Lin et al., 2024). Developing a unified end-to-end analysis that simultaneously accounts for both training and sampling schedules could provide a deeper theoretical foundation, especially since the theoretical impact of discretization schedules on training error remains largely unexplored. For example, how the training-time discretization/noise schedule shapes the time profile of the score error in Assumption 3.1, including both its typical magnitude $\varepsilon_t$ and its state sensitivity $L_t$ across noise levels. In particular, connecting our sampling-side accumulation bound with training-side guarantees would require time-resolved estimates of the score-estimation error at each noise level, which remains an open challenge.

Another natural direction is to study how score errors accumulate under higher-order samplers. The present analysis focuses on first-order sampling methods. In higher-order methods, multi-stage updates and higher-order corrections

may fundamentally influence the amplification mechanism.

Finally, extending our analysis to less benign settings where standard regularity conditions—such as those in Assumption 3.1—do not hold presents a critical theoretical challenge. A particularly important direction is the data manifold regime, where the data distribution is supported on a low-dimensional manifold and the ground-truth score becomes singular in the small-noise limit. Investigating how error propagation behaves in such singular regimes would significantly bridge the gap between our theoretical bounds and complex, real-world data distributions.

## Impact Statement

This paper focuses on the theoretical analysis of score accumulation in diffusion models. Our contributions are methodological and do not introduce new applications with direct societal impact. Therefore, we do not anticipate unique ethical consequences beyond those inherent to generative modeling research.

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

## A. Proof of Main Results

*Proof of Theorem 3.5.* Couple the perturbed and baseline chains synchronously using the same Gaussian noises $\{z_k\}_{k=0}^{K-1}$. Let $\Delta_k := y_k - y_k^{(0)}$ denote the pathwise discrepancy. Subtracting the baseline update from the perturbed update yields:

$$\Delta_{k+1} = \alpha_k \Delta_k + \beta_k \left( \nabla \log p_{\tau_k}(y_k) - \nabla \log p_{\tau_k}(y_k^{(0)}) \right) + \beta_k e(\tau_k, y_k).$$

By the Mean Value Theorem for vector-valued functions, the difference in the score functions can be expressed as:

$$\nabla \log p_{\tau_k}(y_k) - \nabla \log p_{\tau_k}(y_k^{(0)}) = H_k \Delta_k, \quad \text{where } H_k = \int_0^1 \nabla^2 \log p_{\tau_k}\left( y_k^{(0)} + t\Delta_k \right) dt.$$

Under the Gaussian score representation (11), the Hessian is constant, $H_k = -\Sigma_{\tau_k}^{-1}$, which simplifies the integral to the exact Hessian. Similarly, for the error term $e(\tau_k, y_k)$, we decompose it as $e(\tau_k, y_k^{(0)}) + [e(\tau_k, y_k) - e(\tau_k, y_k^{(0)})]$ and apply the Mean Value Theorem to the difference:

$$e(\tau_k, y_k) - e(\tau_k, y_k^{(0)}) = \bar{E}_k \Delta_k, \quad \text{where } \bar{E}_k = \int_0^1 \nabla_x e\left( \tau_k, y_k^{(0)} + t\Delta_k \right) dt.$$

By Assumption 3.1(1), the spectral norm of this average Jacobian is bounded by $\|\bar{E}_k\|_{\text{op}} \leq L_{\tau_k}$. Substituting these into the dynamics, we obtain:

$$\Delta_{k+1} = \underbrace{\left( \alpha_k I_d + \beta_k H_k + \beta_k \bar{E}_k \right)}_{\Phi_k} \Delta_k + \beta_k e(\tau_k, y_k^{(0)}).$$

Iterating this linear recurrence from $\Delta_0 = 0$ to the terminal step $K$:

$$\Delta_K = \sum_{i=0}^{K-1} \left( \prod_{j=i+1}^{K-1} \Phi_j \right) \beta_i e(\tau_i, y_i^{(0)}).$$

Taking the $L_2$ norm and applying the triangle inequality followed by the sub-multiplicativity of the operator norm:

$$(\mathbb{E}\|\Delta_K\|^2)^{1/2} \leq \sum_{i=0}^{K-1} \left( \prod_{j=i+1}^{K-1} \|\Phi_j\|_{\text{op}} \right) \beta_i (\mathbb{E}\|e(\tau_i, y_i^{(0)})\|^2)^{1/2}.$$

Using the triangle inequality on the operator norm and the definition of $\lambda_j$:

$$\|\Phi_j\|_{\text{op}} \leq \|\alpha_j I_d + \beta_j H_j\|_{\text{op}} + \beta_j \|\bar{E}_j\|_{\text{op}} \leq \lambda_j.$$

It follows that the cumulative gain satisfies $\prod \|\Phi_j\|_{\text{op}} \beta_i \leq g_i$. Squaring both sides and applying the Cauchy–Schwarz inequality $(\sum a_i)^2 \leq K \sum a_i^2$ yields:

$$\mathbb{E}\|\Delta_K\|^2 \leq K \sum_{i=0}^{K-1} g_i^2 \varepsilon_{\tau_i}^2.$$

Since $W_2^2(p(y_K), p(y_K^{(0)})) \leq \mathbb{E}\|\Delta_K\|^2$ for the synchronous coupling, the result holds. $\square$

*proof of Theorem 3.7.* Let $\Delta_K := y_K - y_K^{(0)}$ under the synchronous coupling. Similar to the proof of Theorem 3.5, we have the perturbation recursion unrolls as

$$\Delta_K = \sum_{i=0}^{K-1} G_i(H) e_{\tau_i},$$

where we abbreviate $e_{\tau_i} := e(\tau_i, y_i^{(0)})$ for simplicity and $G_i(H) := \beta_i \prod_{j=i+1}^{K-1} \Phi_j$. Add and subtract the surrogate gains $G_i(\bar{H})$:

$$\Delta_K = \underbrace{\sum_{i=0}^{K-1} G_i(\bar{H}) e_{\tau_i}}_{=:S_1} + \underbrace{\sum_{i=0}^{K-1} \left( G_i(H) - G_i(\bar{H}) \right) e_{\tau_i}}_{=:S_2}.$$

Hence
$$W_2^2\big(p_K, p_K^{(0)}\big) = \mathbb{E}\|\Delta_K\|^2 \leq 2\mathbb{E}\|S_1\|^2 + 2\mathbb{E}\|S_2\|^2.$$

*Control of $S_1$.* By the Cauchy–Schwarz inequality $(\sum a_i)^2 \leq K \sum a_i^2$, we have

$$\mathbb{E}\|S_1\|^2 \leq K \sum_{i=0}^{K-1} g_i(\bar{H})^2 \varepsilon_{\tau_i}^2.$$

*Control of $S_2$.* By expanding the product in $G_i(H)$ and using submultiplicativity,

$$G_i(H) - G_i(\bar{H}) = \left( \prod_{\ell=i+1}^{K-1} \big(\alpha_\ell I_d + \beta_\ell(H_\ell + \bar{E}_\ell)\big) - \prod_{\ell=i+1}^{K-1} \big(\alpha_\ell I_d + \beta_\ell(\bar{H}_\ell + \bar{E}_\ell)\big) \right)\beta_i$$

$$= \sum_{j=i+1}^{K-1} \left( \prod_{\ell=j+1}^{K-1} \big(\alpha_\ell I_d + \beta_\ell(H_\ell + \bar{E}_\ell)\big) \right)\beta_j(H_j - \bar{H}_j)\left( \prod_{\ell=i+1}^{j-1} \big(\alpha_\ell I_d + \beta_\ell(\bar{H}_\ell + \bar{E}_\ell)\big) \right)\beta_i.$$

*Taking operator norms and using submultiplicativity:*

$$\|G_i(H) - G_i(\bar{H})\|_{\mathrm{op}} \leq \sum_{j=i+1}^{K-1} \left( \prod_{\ell=j+1}^{K-1} \|\alpha_\ell I_d + \beta_\ell(H_\ell + \bar{E}_\ell)\|_{\mathrm{op}} \right)\beta_j \, \|H_j - \bar{H}_j\|_{\mathrm{op}}$$

$$\times \left( \prod_{\ell=i+1}^{j-1} \|\alpha_\ell I_d + \beta_\ell(\bar{H}_\ell + \bar{E}_\ell)\|_{\mathrm{op}} \right)\beta_i.$$

Assume there exists a constant $C_0 \geq 1$, defined by

$$C_0 := \max_\ell \left\{ 1 + \beta_\ell\big(\Lambda_\ell + L_{\tau_\ell}\big) \right\}, \qquad \Lambda_\ell := \max_m \|\Sigma_m(\tau_\ell)^{-1}\|_{\mathrm{op}},$$

such that for all relevant $\ell$,

$$\|\alpha_\ell I_d + \beta_\ell(H_\ell + \bar{E}_\ell)\|_{\mathrm{op}} \leq C_0, \qquad \|\alpha_\ell I_d + \beta_\ell(\bar{H}_\ell + \bar{E}_\ell)\|_{\mathrm{op}} \leq C_0.$$

Then each product is bounded by a constant that we absorb into $C$, yielding

$$\|G_i(H) - G_i(\bar{H})\|_{\mathrm{op}} \leq C \beta_i \sum_{j=i+1}^{K-1} \beta_j \, \|H_j - \bar{H}_j\|_{\mathrm{op}}.$$

Define
$$\mathcal{S}_0 := \max_{0 \leq i \leq K-1} \epsilon_{\tau_i}$$

Hence

$$\mathbb{E}\|S_2\| = \mathbb{E}\left\| \sum_{i=0}^{K-1} \big(G_i(H) - G_i(\bar{H})\big)e_{\tau_i} \right\|$$

$$\leq \sum_{i=0}^{K-1} \mathbb{E}\|G_i(H) - G_i(\bar{H})\|_{\mathrm{op}} \, \mathbb{E}\|e_{\tau_i}\|$$

$$\leq C \sum_{i=0}^{K-1} \left( \beta_i \sum_{j=i+1}^{K-1} \beta_j \mathbb{E}\|H_j - \bar{H}_j\|_{\mathrm{op}} \right) \mathbb{E}\|e_{\tau_i}\|$$

$$\leq C \left( \sum_{i=0}^{K-1} \beta_i \sum_{j=i+1}^{K-1} \beta_j(d+2)\Lambda_j \right) \mathcal{S}_0$$

The last inequality comes from Lemma C.1: $\mathbb{E}\|H_j - \bar{H}_j\|_{\mathrm{op}} \leq (d+2)\Lambda_j$, we complete the proof. $\qquad\square$

*Proof of Theorem 3.11.* The argument follows the same structure as the proof of Theorem 3.12. In addition, by Lemma C.2 and Lemma C.3, we can control the deviation $\|H_i - \bar{H}_i\|$ depending on the regime of $p_{\tau_i}$: in the *small separation* regime the deviation is $O(\delta(\tau_i))$, while in the *large separation* regime it is $O(e^{-\kappa})$. Combining these bounds with the general estimate in Theorem 3.12 yields inequality (19). $\square$

*Proof of Theorem 3.12.* Let $\Delta_k := y_k - y_k^{(0)}$ under synchronous coupling. Subtracting the baseline update from the perturbed update gives

$$\Delta_{k+1} = a_k \Delta_k + b_k \left( \nabla \log p_{\tau_k}(y_k) - \nabla \log p_{\tau_k}(y_k^{(0)}) \right) + b_k \, e(\tau_k, y_k),$$

with $(a_k, b_k) = (e^{h_k^{\leftarrow}}, 2(e^{h_k^{\leftarrow}} - 1))$ for VP and $(a_k, b_k) = (1, 2h_k^{\leftarrow})$ for VE. By the mean–value representation,

$$\nabla \log p_{\tau_k}(y_k) - \nabla \log p_{\tau_k}(y_k^{(0)}) = H_k \, \Delta_k, \quad H_k := \int_0^1 \nabla^2 \log p_{\tau_k}(y_k^{(0)} + t\Delta_k) \, dt.$$

Lemma B.1 implies

$$\mathbb{E}\|H_k\|_{\mathrm{op}} \leq \frac{d+1}{\sigma^2(\tau_k)} =: C_k.$$

Decomposing $e(\tau_k, y_k)$ as

$$e(\tau_k, y_k) = e(\tau_k, y_k^{(0)}) + \left( e(\tau_k, y_k) - e(\tau_k, y_k^{(0)}) \right)$$

and using Assumption 3.1(1) yields

$$\|e(\tau_k, y_k) - e(\tau_k, y_k^{(0)})\| \leq L_{\tau_k} \|\Delta_k\|.$$

Hence

$$\Delta_{k+1} = \widetilde{M}_k \, \Delta_k + b_k \, e(\tau_k, y_k^{(0)}), \qquad \widetilde{M}_k := a_k I_d + b_k(H_k + L_{\tau_k} I_d),$$

and

$$\mathbb{E}\|\widetilde{M}_k\|_{\mathrm{op}} \leq a_k + b_k(C_k + L_{\tau_k}) =: \alpha_k.$$

Iterating from $\Delta_0 = 0$,

$$\Delta_K = \sum_{i=1}^{K-1} G_i(H) \, e(\tau_i, y_i^{(0)}), \quad G_i(H) := \prod_{j=i+1}^{K-1} \widetilde{M}_j.$$

By Cauchy–Schwarz and Assumption 3.1(ii),

$$\mathbb{E}\|\Delta_K\|^2 \leq K \sum_{i=1}^{K-1} \|G_i(H)\|_{\mathrm{op}}^2 \, \mathbb{E}\|e(\tau_i, y_i^{(0)})\|^2 \leq \sum_{i=1}^{K-1} \|G_i(H)\|_{\mathrm{op}}^2 \varepsilon_{\tau_i}^2.$$

Finally, synchronous coupling gives

$$W_2^2(p_K, p_K^{(0)}) \leq \mathbb{E}\|\Delta_K\|^2,$$

establishing (20). $\square$

## B. Useful Lemmas

**Lemma B.1** (Expected operator–norm Hessian). *Let $X = \mu X_0 + \sigma Z$ with $Z \sim \mathcal{N}(0, I_d)$ independent of an arbitrary $X_0$ in $\mathbb{R}^d$, and let $p_{\mu,\sigma}$ be the density of $X$. Then*

$$\mathbb{E}\left\| \nabla^2 \log p_{\mu,\sigma}(X) \right\|_{\mathrm{op}} \leq \frac{d+1}{\sigma^2}.$$

*Proof of Lemma B.1.* For $X = \mu X_0 + \sigma Z$ with density $p_{\mu,\sigma}$, differentiating the Gaussian-smoothed density under the integral (justified by dominated convergence for the Gaussian kernel) yields, for every $x \in \mathbb{R}^d$,

$$\nabla \log p_{\mu,\sigma}(x) = \frac{1}{\sigma^2}\Big(\mu\mathbb{E}[X_0 \mid X{=}x] - x\Big), \tag{23}$$

$$\nabla^2 \log p_{\mu,\sigma}(x) = \frac{1}{\sigma^4}\mathrm{Cov}(\mu X_0 \mid X{=}x) - \frac{1}{\sigma^2}I_d. \tag{24}$$

From (24) and $\|A\|_{\mathrm{op}} \leq \mathrm{tr}\,(A)$ for $A \succeq 0$,

$$\big\|\nabla^2 \log p_{\mu,\sigma}(x)\big\|_{\mathrm{op}} \leq \frac{1}{\sigma^4}\,\mathrm{tr}\,(\mathrm{Cov}(\mu X_0 \mid X{=}x)) + \frac{1}{\sigma^2}.$$

Taking expectation over $X$ and using the Bayes-risk optimality of the conditional mean,

$$\mathbb{E}\,\mathrm{tr}\,(\mathrm{Cov}(\mu X_0 \mid X))) = \mathbb{E}\mathbb{E}\Big[\big\|\mu X_0 - \mathbb{E}[\mu X_0 \mid X]\big\|^2\Big|X\Big] \leq \mathbb{E}\big\|\mu X_0 - X\big\|^2.$$

Since $X = \mu X_0 + \sigma Z$ with $Z \sim \mathcal{N}(0, I_d)$ independent of $X_0$, we have

$$\mathbb{E}\big\|\mu X_0 - X\big\|^2 = \mathbb{E}\big\| - \sigma Z\big\|^2 = \sigma^2\mathbb{E}\|Z\|^2 = d\sigma^2.$$

Therefore,

$$\mathbb{E}\big\|\nabla^2 \log p_{\mu,\sigma}(X)\big\|_{\mathrm{op}} \leq \frac{1}{\sigma^4}d\sigma^2 + \frac{1}{\sigma^2} = \frac{d+1}{\sigma^2}.$$

$\square$

**Lemma B.2** (Universal expectation bound for Gaussian mixtures). *Let $p_t(x) = \sum_{m=1}^K \pi_m \mathcal{N}\big(x; \mu_m(t), \Sigma_m(t)\big)$ and define*

$$\gamma_m(x) = \frac{\pi_m \varphi_m(x)}{p_t(x)}, \qquad v_m(x) = \Sigma_m(t)^{-1}\big(\mu_m(t) - x\big).$$

*Then, for $X \sim p_t$,*

$$\mathbb{E}\big\|\nabla^2 \log p_t(X)\big\|_{\mathrm{op}} \leq \sum_{m=1}^K \pi_m\big\|\Sigma_m(t)^{-1}\big\|_{\mathrm{op}} + \sum_{m=1}^K \pi_m\,\mathrm{tr}\,\big(\Sigma_m(t)^{-1}\big).$$

*In particular, since $\mathrm{tr}\,(A) \leq d\|A\|_{\mathrm{op}}$,*

$$\mathbb{E}\big\|\nabla^2 \log p_t(X)\big\|_{\mathrm{op}} \leq (d+1)\sum_{m=1}^K \pi_m\big\|\Sigma_m(t)^{-1}\big\|_{\mathrm{op}} \leq (d+1)\max_m\big\|\Sigma_m(t)^{-1}\big\|_{\mathrm{op}}.$$

*Proof of Lemma B.2.* From the mixture Hessian identity,

$$\nabla^2 \log p_t(x) = -\sum_m \gamma_m(x)\Sigma_m(t)^{-1} + \mathrm{Cov}_{m\sim\gamma(\cdot|x)}\big[v_m(x)\big],$$

hence for any $x$,

$$\big\|\nabla^2 \log p_t(x)\big\|_{\mathrm{op}} \leq \Big\|\sum_m \gamma_m(x)\Sigma_m(t)^{-1}\Big\|_{\mathrm{op}} + \mathbb{E}_{\gamma(\cdot|x)}\big\|v_m(x)\big\|^2.$$

*First term.* By triangle inequality, $\|\sum_m \gamma_m(x)\Sigma_m(t)^{-1}\|_{\mathrm{op}} \leq \sum_m \gamma_m(x)\|\Sigma_m(t)^{-1}\|_{\mathrm{op}}$. Taking $\mathbb{E}$ in $X \sim p_t$ and using $\mathbb{E}[\gamma_m(X)] = \pi_m$ gives

$$\mathbb{E}\Big\|\sum_m \gamma_m(X)\Sigma_m(t)^{-1}\Big\|_{\mathrm{op}} \leq \sum_m \pi_m\|\Sigma_m(t)^{-1}\|_{\mathrm{op}}.$$

*Second term.* By the law of total expectation under the generative model $M \sim \{\pi_m\}$, $X|M = m \sim \mathcal{N}(\mu_m(t), \Sigma_m(t))$,

$$\mathbb{E}_X\mathbb{E}_{\gamma(\cdot|X)}\big\|v_m(X)\big\|^2 = \mathbb{E}_{M,X}\big\|\Sigma_M(t)^{-1}\big(\mu_M(t) - X\big)\big\|^2.$$

Condition on $M = m$: $\mu_m(t) - X \sim \mathcal{N}(0, \Sigma_m(t))$, so

$$\mathbb{E}\left[\left\|\Sigma_m(t)^{-1}\big(\mu_m(t) - X\big)\right\|^2 \big| M = m\right] = \operatorname{tr}\left(\Sigma_m(t)^{-1}\right).$$

Averaging over $m$ with weights $\pi_m$ yields $\mathbb{E}_X \mathbb{E}_{\gamma(\cdot|X)}\|v_m(X)\|^2 = \sum_m \pi_m \operatorname{tr}\left(\Sigma_m(t)^{-1}\right)$.

Combine the two bounds to obtain the stated inequality. The final display follows from $\operatorname{tr}(A) \le d\|A\|_{\mathrm{op}}$ and $\sum_m \pi_m a_m \le \max_m a_m$. $\qquad\square$

**Lemma B.3** (Lipschitz score error under Gaussian-mixture marginals). *Assume*

$$p_0(x) = \sum_{k=1}^{K} \pi_k \mathcal{N}(x; m_k, \Sigma_k), \qquad \pi_k > 0, \sum_k \pi_k = 1, \Sigma_k \succ 0.$$

*Let $(p_t)_{t \in (0,T]}$ be the forward marginals of a VP/VE diffusion, so that for each $t > 0$,*

$$p_t(x) = \sum_{k=1}^{K} \pi_k \mathcal{N}\big(x; m_k(t), \Sigma_k(t)\big), \qquad \Sigma_k(t) \succ 0.$$

*Then for every $t \in (0, T]$:*

(a) *The score $\nabla \log p_t(x)$ is globally Lipschitz in $x$, i.e.,*

$$\|\nabla \log p_t(x) - \nabla \log p_t(y)\| \le L_t^\star \|x - y\|, \qquad \forall x, y \in \mathbb{R}^d$$

*for some finite constant $L_t^\star < \infty$.*

(b) *If the learned score $s_\theta(\cdot, t)$ is $L_t^\theta$–Lipschitz in $x$, then the score error*

$$e(t, x) := s_\theta(x, t) - \nabla \log p_t(x)$$

*is $L_t$–Lipschitz with*

$$L_t \le L_t^\theta + L_t^\star.$$

*Proof.* Since $p_0$ is a finite Gaussian mixture, we may write

$$p_0(x) = \sum_{k=1}^{K} \pi_k \mathcal{N}(x; \mu_k(0), \Sigma_k(0)).$$

Under the VP/VE forward dynamics, each component evolves into another Gaussian with mean and covariance given by Eq. (25):

$$p_t(x) = \sum_{k=1}^{K} \pi_k \mathcal{N}\big(x; \mu_k(t), \Sigma_k(t)\big), \qquad t > 0,$$

where

$$\begin{aligned}
\mu_k^{\mathrm{VP}}(t) &= e^{-t}\mu_k(0), & \Sigma_k^{\mathrm{VP}}(t) &= e^{-2t}\Sigma_k(0) + \big(1 - e^{-2t}\big)I_d, \\
\mu_k^{\mathrm{VE}}(t) &= \mu_k(0), & \Sigma_k^{\mathrm{VE}}(t) &= \Sigma_k(0) + 2tI_d.
\end{aligned} \tag{25}$$

For any fixed $t > 0$, all component covariances $\Sigma_k(t)$ are strictly positive definite with eigenvalues uniformly bounded below by a constant $c_t > 0$. Each component density $\varphi_k(x) = \mathcal{N}(x; \mu_k(t), \Sigma_k(t))$ is smooth and strongly log-concave, and its Hessian $\nabla^2 \log \varphi_k(x)$ is a bounded matrix whose operator norm depends only on $\Sigma_k(t)$.

Let

$$p_t(x) = \sum_{k=1}^{K} \pi_k \varphi_k(x), \qquad w_k(x) = \frac{\pi_k \varphi_k(x)}{p_t(x)},$$

so that
$$\nabla \log p_t(x) = \sum_{k=1}^{K} w_k(x) \nabla \log \varphi_k(x).$$

Differentiating,
$$\nabla^2 \log p_t(x) = \sum_{k=1}^{K} w_k(x) \nabla^2 \log \varphi_k(x) + \mathrm{Cov}_{w(x)}\big(\nabla \log \varphi_k(x)\big),$$

where both terms are bounded uniformly in $x$. Hence
$$\sup_{x \in \mathbb{R}^d} \|\nabla^2 \log p_t(x)\|_{\mathrm{op}} < \infty.$$

By the mean-value theorem,
$$\|\nabla \log p_t(x) - \nabla \log p_t(y)\| \leq L_t^\star \|x - y\|,$$

for some finite constant $L_t^\star$ depending only on $t$ and the mixture parameters. This establishes that $\nabla \log p_t$ is globally Lipschitz.

Finally, suppose the learned score $s_\theta(\cdot, t)$ is $L_t^\theta$–Lipschitz in $x$, i.e.,
$$\|s_\theta(x,t) - s_\theta(y,t)\| \leq L_t^\theta \|x - y\|, \qquad \forall\, x, y.$$

Define the score error $e(t,x) = s_\theta(x,t) - \nabla \log p_t(x)$. Then for any $x, y$,
$$\|e(t,x) - e(t,y)\| \leq \|s_\theta(x,t) - s_\theta(y,t)\| + \|\nabla \log p_t(x) - \nabla \log p_t(y)\|.$$

Using the Lipschitz constant $L_t^\star$ established above for $\nabla \log p_t$, we obtain
$$\|e(t,x) - e(t,y)\| \leq (L_t^\theta + L_t^\star)\, \|x - y\|.$$

Thus $e(t, \cdot)$ is $L_t$–Lipschitz with
$$L_t \;\leq\; L_t^\theta + L_t^\star,$$

completing the proof. $\qquad\square$

To further sharpen the behavior of the general bound in Theorem 3.12 as $t \to 0$, we analyze the case where the data distribution $p_0$ has a Lipschitz score function. Under this additional smoothness, the next lemma (Lemma B.4) shows that the operator norm of the Hessian of $\log p_t$ remains uniformly controlled by that of $\log p_0$, and in particular does not exhibit the $1/t$ blow-up present in the distribution-free bound.

**Lemma B.4** (Bounded Hessian expectation for smoothed densities). *Let $p_0$ be a density on $\mathbb{R}^d$ and set $\ell_0 := \log p_0$. Assume:*

*(A1) $p_0(x) > 0$ for all $x$ and $\ell_0 \in C^2(\mathbb{R}^d)$;*

*(A2) the score $\nabla \ell_0$ is globally $L$-Lipschitz, i.e. $\|\nabla^2 \ell_0(x)\|_{\mathrm{op}} \leq L$ for all $x$;*

*(A3) $X_0 \sim p_0$ has finite second moment $\mathbb{E}\|X_0\|^2 \leq M_2 < \infty$.*

*Consider*
$$X_t = \mu_t X_0 + \sigma_t Z, \qquad Z \sim \mathcal{N}(0, I_d) \text{ independent of } X_0,$$

*where $\sigma_t \to 0$ and $\mu_t \to 1$ as $t \to 0$, and let $p_t$ be the density of $X_t$. Then there exist $t_0 > 0$ and $C < \infty$ such that*
$$\sup_{0 < t \leq t_0} \mathbb{E}\big\|\nabla^2 \log p_t(X_t)\big\|_{\mathrm{op}} \leq C.$$

*More precisely, if $\mu_t \geq m > 0$ for all $t \in (0, t_0]$, one may take*
$$C = \frac{L + \big(\|\nabla \ell_0(0)\| + L\sqrt{M_2}\big)^2}{m^2},$$

*where $\|\nabla \ell_0(0)\| < \infty$ since $\ell_0 \in C^2(\mathbb{R}^d)$.*

*Proof of Lemma B.4.* Let $\tilde{\sigma}_t := \sigma_t/\mu_t$ and define $Y_t := X_0 + \tilde{\sigma}_t Z$, so that $X_t = \mu_t Y_t$. Denote by $q_t$ the density of $Y_t$. Since $Y_t$ is obtained by adding Gaussian noise to $X_0$, we have $q_t = p_0 * \phi_{\tilde{\sigma}_t}$, where $\phi_{\tilde{\sigma}_t}$ is the centered Gaussian density with covariance $\tilde{\sigma}_t^2 I_d$.

We will use a standard Gaussian-smoothing identity. Fix $\sigma > 0$, let $Y := X_0 + \sigma Z$ and denote $q_\sigma = p_0 * \phi_\sigma$. Then for every $y \in \mathbb{R}^d$,

$$\nabla^2 \log q_\sigma(y) = \mathbb{E}\big[\nabla^2 \ell_0(X_0) \mid Y = y\big] + \mathrm{Cov}\big(\nabla \ell_0(X_0) \mid Y = y\big), \tag{26}$$

where the conditional expectation and covariance are taken with respect to the posterior law of $X_0$ given $Y = y$.

The first term in (26) is controlled directly by the global Hessian bound in the Lipschitz assumption:

$$\Big\| \mathbb{E}\big[\nabla^2 \ell_0(X_0) \mid Y = y\big] \Big\|_{\mathrm{op}} \leq \mathbb{E}\big[\|\nabla^2 \ell_0(X_0)\|_{\mathrm{op}} \mid Y = y\big] \leq L.$$

For the second term, note that for any random vector $U$, $\mathrm{Cov}(U \mid Y = y) \preceq \mathbb{E}[UU^\top \mid Y = y]$, hence

$$\big\| \mathrm{Cov}(U \mid Y = y) \big\|_{\mathrm{op}} \leq \big\| \mathbb{E}[UU^\top \mid Y = y] \big\|_{\mathrm{op}} \leq \mathbb{E}[\|U\|^2 \mid Y = y].$$

Applying this with $U = \nabla \ell_0(X_0)$ and combining with (26) yields

$$\|\nabla^2 \log q_\sigma(y)\|_{\mathrm{op}} \leq L + \mathbb{E}\big[\|\nabla \ell_0(X_0)\|^2 \mid Y = y\big].$$

Integrating both sides against the density of $Y$ and using the tower property gives

$$\mathbb{E}\|\nabla^2 \log q_\sigma(Y)\|_{\mathrm{op}} \leq L + \mathbb{E}\|\nabla \ell_0(X_0)\|^2.$$

It remains to bound $\mathbb{E}\|\nabla \ell_0(X_0)\|^2$ using the second moment of $X_0$. Since $\nabla \ell_0$ is $L$-Lipschitz,

$$\|\nabla \ell_0(x)\| \leq \|\nabla \ell_0(0)\| + L\|x\|,$$

and therefore, by Cauchy–Schwarz,

$$\mathbb{E}\|\nabla \ell_0(X_0)\|^2 \leq \big(\|\nabla \ell_0(0)\| + L\sqrt{\mathbb{E}\|X_0\|^2}\big)^2 \leq \big(\|\nabla \ell_0(0)\| + L\sqrt{M_2}\big)^2.$$

Substituting this bound into the previous display and specializing to $\sigma = \tilde{\sigma}_t$ yields

$$\mathbb{E}\|\nabla^2 \log q_t(Y_t)\|_{\mathrm{op}} \leq L + \big(\|\nabla \ell_0(0)\| + L\sqrt{M_2}\big)^2.$$

Finally, we transfer the estimate from $q_t$ to $p_t$ using the scaling relation induced by $X_t = \mu_t Y_t$. Indeed, $p_t(x) = \mu_t^{-d} q_t(x/\mu_t)$, and differentiating twice gives

$$\nabla^2 \log p_t(x) = \mu_t^{-2} \nabla^2 \log q_t(x/\mu_t).$$

Evaluating at $x = X_t = \mu_t Y_t$ and taking expectations,

$$\mathbb{E}\|\nabla^2 \log p_t(X_t)\|_{\mathrm{op}} = \mu_t^{-2}\mathbb{E}\|\nabla^2 \log q_t(Y_t)\|_{\mathrm{op}} \leq \mu_t^{-2}\Big(L + \big(\|\nabla \ell_0(0)\| + L\sqrt{M_2}\big)^2\Big).$$

Since $\mu_t \to 1$ as $t \to 0$, we can choose $t_0 > 0$ and $m > 0$ such that $\mu_t \geq m$ for all $t \in (0, t_0]$, which implies

$$\sup_{0 < t \leq t_0} \mathbb{E}\|\nabla^2 \log p_t(X_t)\|_{\mathrm{op}} \leq \frac{L + \big(\|\nabla \ell_0(0)\| + L\sqrt{M_2}\big)^2}{m^2}.$$

This completes the proof. $\square$

## C. Gaussian Mixture Hessian Approximation

**Hessian decomposition and responsibilities.** For Gaussian mixtures

$$p_t(x) = \sum_{m=1}^{K} \pi_m \mathcal{N}(x; \mu_m(t), \Sigma_m(t)),$$

the (posterior) responsibility of component $m$ at location $x$ is

$$\gamma_m(x) := \frac{\pi_m \mathcal{N}(x; \mu_m(t), \Sigma_m(t))}{\sum_{\ell=1}^{K} \pi_\ell \mathcal{N}(x; \mu_\ell(t), \Sigma_\ell(t))}.$$

With this notation, the score Hessian admits the exact decomposition

$$\nabla^2 \log p_t(x) = -\sum_{m=1}^{K} \gamma_m(x) \Sigma_m(t)^{-1} + \mathrm{Cov}_{m \sim \gamma(\cdot|x)}[v_m(x)], \quad v_m(x) := \Sigma_m(t)^{-1}(\mu_m(t) - x). \quad (27)$$

**Separation regimes.** Define the mean–separation surrogate

$$\delta_\mu(t) := \max_{m \neq n} \left\| \Sigma_m(t)^{-1/2} \big( \mu_m(t) - \mu_n(t) \big) \right\|,$$

and the covariance–separation surrogate

$$\delta_\Sigma(t) := \max_{m,n} \left\| \Sigma_m(t)^{1/2} \Sigma_n(t)^{-1} \Sigma_m(t)^{1/2} - I_d \right\|_{\mathrm{op}}.$$

We bundle them into a single small–separation parameter

$$\delta(t) := \max \big( \delta_\mu(t), \delta_\Sigma(t) \big).$$

We say *small separation* if $\delta(t) \ll 1$.

For large separation, define the logit margin

$$\kappa_t(x) := \min_{j \neq i^*(x)} \big( \ell_{i^*}(x) - \ell_j(x) \big), \qquad i^*(x) = \arg\max_i \ell_i(x),$$

where $\ell_i(x) = \log \pi_i - \frac{1}{2} \log \det(2\pi \Sigma_i(t)) - \frac{1}{2}(x - \mu_i(t))^\top \Sigma_i(t)^{-1}(x - \mu_i(t))$. We say *large separation* along a path $\{x_t\}$ if $\kappa_{\tau_k}(x_t) \geq \underline{\kappa} \gg 1$ for all $t \in [0,1]$.

The mixture Hessian can be approximated by surrogates of the form $-\sum_m w_m \Sigma_m^{-1}$ with different choices of weights $w_m$. A crude bound is always available by taking the prior weights $\pi_m$, but this ignores how the posterior responsibilities $\gamma_m(x)$ behave in different regimes. In the small–separation regime, the responsibilities remain close to the prior $\pi$, so the surrogate $\bar{H}_k = -\sum_m \pi_m \Sigma_m^{-1}$ achieves accuracy $O(\Lambda \delta(\tau_k))$. In the large–separation regime, the posterior mass concentrates sharply on one component, so a hard surrogate $\bar{H}_k = -\Sigma_{i^*}^{-1}$ is more appropriate, leading to exponential accuracy $O(\Lambda e^{-\underline{\kappa}})$. Accordingly, we analyze these three cases separately: a crude uniform bound (Lemma C.1), a refined small–separation bound (Lemma C.2), and a large–separation bound (Lemma C.3).

**Lemma C.1** (Crude uniform bound for surrogate Hessians). *Let*

$$H_k = \int_0^1 \nabla^2 \log p_{\tau_k} \big( y_k^{(0)} + t\Delta_k \big) \, dt, \qquad \bar{H}_k \in \left\{ -\sum_{m=1}^{K} \pi_m \Sigma_m(\tau_k)^{-1}, \ -\Sigma_{i^\star(y_k)}(\tau_k)^{-1} \right\},$$

*where $i^\star(y_k) = \arg\max_{m \in [K]} \gamma_m(y_k; \tau_k)$, and set $\Lambda := \max_{m \in [K]} \|\Sigma_m(\tau_k)^{-1}\|_{\mathrm{op}}$. Then*

$$\mathbb{E} \big\| H_k - \bar{H}_k \big\|_{\mathrm{op}} \leq (d+2) \Lambda. \quad (28)$$

*Proof.* By the mixture Hessian identity,

$$\nabla^2 \log p_{\tau_k}(x) = -\sum_{m=1}^{K} \gamma_m(x; \tau_k) \, \Sigma_m(\tau_k)^{-1} + \mathrm{Cov}_{m\sim\gamma(\cdot|x;\tau_k)}\big[\Sigma_m(\tau_k)^{-1}(\mu_m(\tau_k) - x)\big].$$

Averaging along the segment $x_t := y_k^{(0)} + t\Delta_k$ and subtracting $\bar{H}_k$ gives

$$H_k - \bar{H}_k = \underbrace{-\int_0^1 \sum_{m=1}^{K} w_m(x_t) \, \Sigma_m(\tau_k)^{-1} \, dt}_{\text{term 1}} + \underbrace{\int_0^1 \mathrm{Cov}_{m\sim\gamma(\cdot|x_t;\tau_k)}[v_m(x_t)] \, dt}_{\text{term 2}},$$

where $v_m(x) = \Sigma_m(\tau_k)^{-1}(\mu_m(\tau_k) - x)$ and

$$w_m(x_t) = \begin{cases} \gamma_m(x_t; \tau_k) - \pi_m, & \text{if } \bar{H}_k = -\sum_j \pi_j \Sigma_j(\tau_k)^{-1}, \\ \gamma_m(x_t; \tau_k) - \mathbf{1}_{\{m=i^\star(y_k)\}}, & \text{if } \bar{H}_k = -\Sigma_{i^\star(y_k)}(\tau_k)^{-1}. \end{cases}$$

**Term 1** For any choice of $w_m$ above,

$$\Big\| \sum_{m=1}^{K} w_m(x_t) \, \Sigma_m(\tau_k)^{-1} \Big\|_{\mathrm{op}} \leq \sum_{m=1}^{K} |w_m(x_t)| \, \|\Sigma_m(\tau_k)^{-1}\|_{\mathrm{op}} \leq \|w(x_t)\|_1 \, \Lambda.$$

In the mixture-weighted case, $\|w(x_t)\|_1 = \|\gamma(x_t; \tau_k) - \pi\|_1 \leq 2$.

In case where $\bar{H}_k = -\Sigma_{i^\star(y_k)}(\tau_k)^{-1}$, writing $m^\star = i^\star(y_k)$,

$$\|w(x_t)\|_1 = \sum_m |\gamma_m(x_t; \tau_k) - \mathbf{1}_{\{m=m^\star\}}| = 2\big(1 - \gamma_{m^\star}(x_t; \tau_k)\big) \leq 2.$$

Thus $\mathbb{E}\|\text{term 1}\| \leq 2\Lambda$.

**Term 2** Since covariance is PSD and $\|A\|_{\mathrm{op}} \leq \mathrm{tr}(A))$,

$$\big\| \mathrm{Cov}_{m\sim\gamma(\cdot|x)}[v_m(x)] \big\|_{\mathrm{op}} \leq \sum_{m=1}^{K} \gamma_m(x; \tau_k) \, \mathrm{tr}\big(\Sigma_m(\tau_k)^{-1}\big) \leq d\,\Lambda.$$

Integrating over $t \in [0, 1]$ and taking expectation obtains $\mathbb{E}\|\text{term 2}\| \leq d\,\Lambda$.

Combining the bounds for the two terms gets $\mathbb{E}\|H_k - \bar{H}_k\|_{\mathrm{op}} \leq d\,\Lambda + 2\,\Lambda = (d+2)\Lambda$, which proves (28). □

**Lemma C.2** (Small separation bound). *In the setting of Lemma C.1, assume the small–separation condition $\delta(\tau_k) = \max(\delta_\mu(\tau_k), \delta_\Sigma(\tau_k)) \ll 1$. Then*

$$\mathbb{E}\|H_k - \bar{H}_k\|_{\mathrm{op}} = O\big(\Lambda\,\delta(\tau_k)\big). \tag{29}$$

*Proof.* We have the decomposition

$$H_k - \bar{H}_k = \underbrace{-\int_0^1 \sum_m (\gamma_m(x_t) - \pi_m) \, \Sigma_m(\tau_k)^{-1} \, dt}_{\text{term 1}} + \underbrace{\int_0^1 \mathrm{Cov}_{m\sim\gamma(\cdot|x_t)}[v_m(x_t)] \, dt}_{\text{term 2}},$$

**Term 1.** Define the logits

$$\theta_m(x) := \log \pi_m - \tfrac{1}{2} \log \det(2\pi\Sigma_m) - \tfrac{1}{2}(x - \mu_m)^\top \Sigma_m^{-1}(x - \mu_m), \quad m = 1, \ldots, K,$$

where $\pi = (\pi_1, \ldots, \pi_K)$ are the mixture weights with $\pi_m > 0$ and $\sum_{m=1}^K \pi_m = 1$. Let $\gamma(x) = (\gamma_1(x), \ldots, \gamma_K(x))$ denote the posterior component weights ("responsibilities") at $x$. Then

$$\gamma(x) = \mathrm{softmax}(\theta(x)), \qquad \pi = \mathrm{softmax}(\theta^0), \quad \theta_m^0 := \log \pi_m.$$

The Jacobian of the softmax map is $J(\theta) = \mathrm{Diag}(\gamma) - \gamma\gamma^\top$, which satisfies $\|J(\theta)\|_{\mathrm{op}} \le \frac{1}{2}$. By the mean value theorem,

$$\|\gamma(x) - \pi\|_2 \le \tfrac{1}{2}\|\theta(x) - \theta^0\|_2.$$

Consequently,

$$\sum_{m=1}^K |\gamma_m(x) - \pi_m| = \|\gamma(x) - \pi\|_1 \le \frac{\sqrt{K}}{2}\,\|\theta(x) - \theta^0\|_2.$$

Now, when $\delta(\tau_k) \ll 1$, the mixture parameters $(\mu_m, \Sigma_m)$ are close to some average $(\bar\mu, \bar\Sigma)$. Writing $z = \bar\Sigma^{-1/2}(x - \bar\mu)$, a Taylor expansion shows

$$|\theta_m(x) - \theta_m^0| \;\le\; C\,\delta(\tau_k)\,\big(1 + \|z\|^2\big).$$

Hence

$$\sum_{m=1}^K |\gamma_m(x) - \pi_m| \;\le\; C\,\delta(\tau_k)\,\big(1 + \|z\|^2\big).$$

Taking expectations gives the desired control:

$$\mathbb{E}\|\text{term 1}\| \;\le\; O\big(\Lambda\,\delta(\tau_k)\big).$$

**Term 2**  Fix $x$ and define, as above,

$$\mathrm{Cov}_{m\sim\gamma(\cdot|x)}[v_m(x)] = \mathbb{E}_{m\sim\gamma(\cdot|x)}\big[(v_m(x) - \bar v(x))(v_m(x) - \bar v(x))^\top\big], \qquad \bar v(x) = \mathbb{E}_{m\sim\gamma(\cdot|x)}v_m(x).$$

By PSD and $\|A\|_{\mathrm{op}} \le \mathrm{tr}(A)$,

$$\big\|\mathrm{Cov}_{m\sim\gamma(\cdot|x)}[v_m(x)]\big\|_{\mathrm{op}} \le \mathbb{E}_{m\sim\gamma(\cdot|x)}\|v_m(x) - \bar v(x)\|^2.$$

Again using the small–separation condition and the same $z = \bar\Sigma(\tau_k)^{-1/2}(x - \bar\mu(\tau_k))$, one has the component spread bound

$$v_m(x) - v_n(x) = \Sigma_m(\tau_k)^{-1}\big(\mu_m(\tau_k) - x\big) - \Sigma_n(\tau_k)^{-1}\big(\mu_n(\tau_k) - x\big)$$
$$= \big(\Sigma_m(\tau_k)^{-1} - \Sigma_n(\tau_k)^{-1}\big)\big(\mu_m(\tau_k) - x\big) \;+\; \Sigma_n(\tau_k)^{-1}\big(\mu_m(\tau_k) - \mu_n(\tau_k)\big),$$

then

$$\|v_m(x) - v_n(x)\| \le \|\Sigma_m(\tau_k)^{-1} - \Sigma_n(\tau_k)^{-1}\|_{\mathrm{op}}\,\|\mu_m(\tau_k) - x\| \;+\; \|\Sigma_n(\tau_k)^{-1}\|_{\mathrm{op}}\,\|\mu_m(\tau_k) - \mu_n(\tau_k)\|,$$

which implies

$$\big\|\mathrm{Cov}_{m\sim\gamma(\cdot|x)}[v_m(x)]\big\|_{\mathrm{op}} \le C^2\,\Lambda^2\,\delta(\tau_k)^2\,\big(1 + \|z\|^2\big).$$

Averaging over $x$ (hence $z$) and $t \in [0,1]$, and using $\mathbb{E}(1 + \|z\|^2) = O(1)$, we get

$$\mathbb{E}\|\text{term 2}\| \;\le\; O\big(\Lambda\,\delta(\tau_k)^2\big).$$

Together with term 1, this yields (29). □

**Lemma C.3** (Large separation with hard surrogate). *Let*

$$H_k = \int_0^1 \nabla^2 \log p_{\tau_k}(x_t)\,dt, \qquad \bar H_k = \bar H_k^{\mathrm{hard}} = -\Sigma_{i^\star(y_k)}(\tau_k)^{-1},$$

*where $x_t = y_k^{(0)} + t\Delta_k$ and $i^\star(x) = \arg\max_m \ell_m(x)$. Assume a uniform logit margin $\kappa_{\tau_k}(x_t) \ge \underline\kappa \gg 1$ for all $t \in [0,1]$. Then*

$$\mathbb{E}\|H_k - \bar H_k^{\mathrm{hard}}\|_{\mathrm{op}} = O\big(\Lambda\,e^{-\underline\kappa}\big). \tag{30}$$

*Proof.* We have the decomposition

$$H_k - \bar{H}_k^{\text{hard}} = \underbrace{-\int_0^1 \left( \sum_m \gamma_m(x_t)\Sigma_m^{-1} - \Sigma_{i^\star(y_k)}^{-1} \right) dt}_{\text{term 1}} + \underbrace{\int_0^1 \text{Cov}_{m\sim\gamma(\cdot|x_t)}[v_m(x_t)]dt}_{\text{term 2}},$$

where $v_m(x) = \Sigma_m^{-1}(\mu_m - x)$ and $\Lambda := \max_m \|\Sigma_m^{-1}\|_{\text{op}}$.

**Term 1**   Insert and subtract $\Sigma_{i^\star(x_t)}^{-1}$:

$$\sum_m \gamma_m(x_t)\Sigma_m^{-1} - \Sigma_{i^\star(y_k)}^{-1} = \sum_{m\neq i^\star(x_t)} \gamma_m(x_t)(\Sigma_m^{-1} - \Sigma_{i^\star(x_t)}^{-1}) + (\Sigma_{i^\star(x_t)}^{-1} - \Sigma_{i^\star(y_k)}^{-1}).$$

The first bracket is bounded by

$$\Big\| \sum_{m\neq i^\star(x_t)} \gamma_m(x_t)(\Sigma_m^{-1} - \Sigma_{i^\star(x_t)}^{-1}) \Big\|_{\text{op}} \leq 2\Lambda \sum_{m\neq i^\star(x_t)} \gamma_m(x_t).$$

The uniform margin implies

$$\sum_{m\neq i^\star(x_t)} \gamma_m(x_t) \leq Ce^{-\underline{\kappa}}. \tag{31}$$

The index mismatch contributes at most $2\Lambda Ce^{-\underline{\kappa}}$. Hence

$$\mathbb{E}\|\text{term 1}\| \ \leq \ C_1\Lambda e^{-\underline{\kappa}}.$$

**Term 2**   Expanding around $i^\star(x_t)$,

$$\|\text{Cov}_{m\sim\gamma(\cdot|x_t)}[v_m(x_t)]\|_{\text{op}} \leq \sum_{m\neq i^\star(x_t)} \gamma_m(x_t)\|v_m(x_t) - v_{i^\star(x_t)}(x_t)\|^2.$$

Since $\|v_m - v_{i^\star}\|^2 = O(\Lambda)$ under bounded moments, and apply (31), we obtain

$$\mathbb{E}\|\text{term 2}\| \ \leq \ C_2\Lambda e^{-\underline{\kappa}}.$$

Adding both terms gives (30). $\qquad\square$

