# OpenReview forum: "The Accumulation of Score Estimation Error in Diffusion Models"
_ICML.cc/2026/Conference — ICML 2026 regular_

### Official Review · Reviewer_SyHn · 2026-02-27

**Soundness:** 4
**Presentation:** 3
**Significance:** 3
**Originality:** 3
**Overall Recommendation:** 4
**Confidence:** 3

**Summary:**

This paper analyzes the accumulation of score estimation errors through the reverse sampling process. Prior work bounds the effect of score error as  $\sim T\epsilon^2$, treating all timesteps equally. This paper provides per-step analysis accounting for the curvature at each step, showing that errors near the data end are amplified more than errors near the noise end. Results are derived for Gaussian distributions (exact Hessian, sharp bound), Gaussian mixtures, and general distributions (conservative bound). They cover both VP and VE diffusions, and both reverse SDE and PF-ODE samplers.

**Compliance With Llm Reviewing Policy:**

Affirmed.

**Final Justification:**

The rebuttal was helpful. I will maintain my positive score.

**Key Questions For Authors:**

Could the bounds be used to help improve the step-size schedule?

**Limitations:**

Yes

**Strengths And Weaknesses:**

Strengths:
  - Nice theoretical contribution: incorporating curvature into the error analysis is a
  natural and helpful refinement that clarifies previous empirical observations that step-size schedules emphasizing the data end are helpful.
  - The progression from Gaussian to Gaussian mixture to general distributions is
  well-structured and clear.
  - The framework is general enough to cover VP/VE
  and SDE/ODE.

  Weaknesses:
  - The practical insight (use smaller steps near the data end) was already known
  empirically. The contribution is formalizing why, which is appreciated but limits the
  significance.
  - Experiments are limited to a Gaussian mixture. This is enough to validate the
  theoretical bounds, but experiments on more realistic settings (e.g., pretrained models on image
   data) would strengthen the practical contribution.
  - The general-distribution bound (Thm 3.12) is conservative, as the authors acknowledge.
  - Possible typo: in the proof of Theorem 3.12, should the sum start from $i=0$ instead of $i=1$ to be consistent with the theorem statement?
  - Also, please see Questions.

---

> ### Author Rebuttal · Authors · 2026-03-31
>
> We thank the reviewer for the valuable comments and questions. Below we provide responses to each point.
>
> ## Significance and practical insight
>
> Even though the practical preference for taking smaller steps near the data end was already observed empirically, our contribution is not to claim this intuition as a new empirical discovery, but to provide a theoretical explanation for it. In particular, our analysis makes explicit how local score-estimation errors are injected and then amplified along the reverse trajectory, thereby explaining why the data-end region is especially sensitive and why schedules with finer refinement there are more robust.
>
> In this sense, the novelty lies in formalizing the mechanism behind an observed phenomenon, rather than in proposing a new heuristic itself. We will revise the manuscript to make this positioning clearer.
>
> ## Realistic experiments
>
>  We have added a new CIFAR-10 experiment in the revision. In this experiment, a subset of CIFAR-10 is treated as an empirical distribution. Under Gaussian noising, its perturbed marginal becomes a finite Gaussian mixture, where we can use the exact empirical score for all $t>0$. We then run a controlled experiment in which, over the last $K$ reverse steps near the data end, we replace the exact empirical score by a learned score, while keeping the exact empirical score for the remaining earlier steps. We evaluate the final discrepancy using sliced Wasserstein distance.
>
> The results show the same qualitative trend as in Figure 1: schedules with stronger refinement near the data end are consistently more robust to injected score error. In particular, uniform log-SNR performs best across all tested settings, while the linear schedule performs worst. The results are as follows:
>
> | Cutoff $K$ | Linear | Quadratic | Cosine | Uniform log-SNR |
> |---:|---:|---:|---:|---:|
> | 100 | 0.2284 | 0.0617 | 0.0869 | **0.0401** |
> | 200 | 0.6858 | 0.1657 | 0.2329 | **0.0461** |
> | 300 | 1.4101 | 0.3866 | 0.4289 | **0.0912** |
> | 400 | 2.5693 | 0.7511 | 0.7105 | **0.3811** |
>
> We will include these results in the revision to demonstrate that the mechanism identified by our theory is not limited to low-dimensional synthetic mixtures, but also appears on real image data.
>
> ## General-distribution bound
>
>
> Theorem 3.12 is intended as a worst-case baseline without structural assumptions on $p_0$, so it is naturally conservative. Its role is to show that, even in the fully distribution-free setting, the data-end region is inherently more sensitive to score-estimation error. As discussed in Corollary 3.14, once additional regularity is imposed, this worst-case bound can be refined and the small-time blow-up can be improved. We will revise the manuscript to make this positioning clearer.
>
> ## Typo in Theorem 3.12 proof
> Thank you for catching this. We will correct it in the revised version.
>
> ## Schedule design
>
> To design an optimal step-size schedule directly from our bounds, one would need precise knowledge of the score-estimation error profile over time, such as $\varepsilon_t$ and $L_t$, which is generally unavailable in practice. For this reason, our bounds are mainly intended to provide a qualitative understanding of the error-accumulation mechanism, rather than to serve as a direct objective for optimization. At the same time, this understanding is still informative for schedule design. By showing that error amplification becomes much stronger near the data end, our framework helps explain why schedules that allocate more steps at small $t$, such as uniform log-SNR, are more robust than linear schedules. While our analysis does not yet produce an explicit optimal schedule, it does provide a principled guide for designing better heuristic schedules.

---

> > ### Author Rebuttal · Reviewer_SyHn · 2026-04-02
> >
> > Thank you for the clarifications, and I think the CIFAR-10 experiment is a nice addition. I will keep my positive score.

---

> > > ### Author Response · Authors · 2026-04-07
> > >
> > > Thank you very much for your careful reading and thoughtful comments. We are glad that the points you raised have been addressed. We sincerely appreciate your time and valuable feedback.

---

### Official Review · Reviewer_6PjJ · 2026-03-04

**Soundness:** 3
**Presentation:** 3
**Significance:** 2
**Originality:** 2
**Overall Recommendation:** 4
**Confidence:** 3

**Summary:**

The paper analyzes how score estimation errors in diffusion models propagate during sampling. It derives non-asymptotic Wasserstein-2 bounds between sampling based on an approximate score function and the ideal reverse-time process, with especially explicit results for Gaussian and Gaussian-mixture settings and a looser bound for general distributions. The bounds indicate higher sensitivity near the data end, motivating smaller step sizes there, and suggest that SDE sampling can be more robust to score error than probability-flow ODE sampling.

**Compliance With Llm Reviewing Policy:**

Affirmed.

**Final Justification:**

The authors have sufficiently addressed my concerns. I find the paper technically solid and the theoretical contribution meaningful in explaining error accumulation in diffusion sampling. I will therefore raise my score to weak accept.

**Key Questions For Authors:**

- In Assumption 3.1, the Lipschitz constant $L_t$ may become large as $t\to 0$. Do the authors expect $L_t$ to blow up near the data end in practice, and is there a relationship between $L_t$ and the constant $C_0$ in Corollary 3.14?
- Could the authors include a direct experiment comparing SDE vs PF-ODE sampling to test whether the predicted difference in score error accumulation is observable in practice?
- How do the authors expect discretization error to interact with score estimation error? Can the framework be extended to control both jointly and obtain a bound on the total sampling error?
- Beyond Gaussian mixtures, can the authors validate the theoretical results on at least one standard diffusion benchmark?

**Limitations:**

Yes

**Strengths And Weaknesses:**

**Strengths**

- The paper is easy to follow: the setup and theoretical results are presented clearly.
- It provides a clear theoretical analysis of score estimation error as a distinct source of sampling error in diffusion models, whereas much of the existing literature focuses on discretization error.
- The theory is developed progressively: it starts from tractable settings (Gaussian and Gaussian mixtures) where more closed-form results are possible, and then extends the analysis to general distributions, with correspondingly looser bounds.


**Weaknesses**

- The theoretical results are still largely in Gaussian / Gaussian-mixture settings and it remains unclear how the theory is for general distributions.
- In Assumption 3.1, the Lipschitz constant $L_t$ may blow up as $t \to 0$. Likewise, the log-smoothness constant $C_0$ in Corollary 3.14 may be large in practice. This can lead to loose bounds near the data end and limits interpretability in typical regimes.
- The claim that SDE sampling amplifies score-estimation error less than PF-ODE sampling is derived mainly in a Gaussian/small-step regime and at the level of upper bounds. Moreover, discretization error, which can be a major contributor to sampling error in practice, is not analyzed, so the comparison does not directly translate to the sample quality.
- Empirical validation is somewhat limited: experiments are restricted to synthetic Gaussian-mixture setups, and there are no evaluations on standard deep-learning benchmarks such as image datasets.

---

> ### Author Rebuttal · Authors · 2026-03-31
>
> We thank the reviewer for the detailed and helpful comments and questions. Below we provide responses to each point.
> ## General distributions vs. Gaussian / Gaussian-mixture settings
>
> Our paper does include a general-distribution result in Theorem 3.12, which provides a distribution-free worst-case bound. This bound is intentionally conservative because it does not assume additional structure on $p_0$. At the same time, our Gaussian-mixture analysis gives a more structured refinement that is also relevant in practice. If a dataset is viewed as a finite empirical distribution, then for every fixed $t>0$, its noisy marginal is exactly a finite Gaussian mixture. In particular, under early stopping at time $\tau>0$, the relevant marginals on $[\tau,T]$ fall into the Gaussian-mixture regime studied in our paper. We will revise the manuscript to make this connection clearer.
>
>
> ## Assumption 3.1, $L_t$, and $C_0$
>
> Here $L_t$ is the Lipschitz constant of the score approximation error $e(x,t)=s_\theta(x,t)-\nabla\log p_t(x)$, while $C_0$ in Corollary 3.14 controls the curvature of the true smoothed density. Thus, they are different quantities. We do expect $L_t$ to become larger near the data end in practice, which is exactly the motivation for introducing the monotonic profile $L_t$ in Assumption 3.1.
>
> If $p_0$ satisfies additional regularity such as a globally Lipschitz score, then the worst-case $1/\sigma(t)^2$ blow-up in Theorem 3.12 can be replaced by a finite constant $C_0$. However, finite does not necessarily mean small, so the resulting bound can still remain conservative in practice. Even so, the underlying error-accumulation mechanism is the same as that illustrated more explicitly in our Gaussian and Gaussian-mixture analyses.
>
> ## SDE vs. PF-ODE comparison
>
> We would like to clarify that we do not claim SDE sampling amplifies score-estimation error less than PF-ODE sampling. Corollary 3.16 and Remark 3.17 only show that the same framework extends to PF-ODE, with the same structural decomposition and different amplification coefficients in $g_i(H)$.
>
> We also conducted an additional comparison between SDE and PF-ODE under the same synthetic score-error conditions. We focus on the large-$K$ regime because it is most aligned with our small-step analysis and reduces the effect of discretization error. In this regime, the accumulated score-estimation errors of ODE and SDE are comparable, and the empirical curves of both samplers closely follow their respective theoretical predictions. We will include these results in the revised appendix.
>
> ## Discretization error and total sampling error
>
> We expect discretization error and score-estimation error to interact through their joint propagation along the reverse-time dynamics. A natural extension of our framework is therefore to augment the current gain recursion with an additional local truncation term $r_k$, leading schematically to
> $$
> \Delta_{k+1}=
> \lambda_k \Delta_k
> +\beta_k e(y_k^{(0)},\tau_k)
> +r_k.
> $$
> Unrolling this recursion would yield a bound in which both score-estimation and discretization errors contribute to the total sampling error through the same gain factors.
>
> We do not claim this joint result in the current paper, since it requires stronger regularity than our present analysis. In particular, one would need sharp control of the local truncation residual $r_k$, which depends not only on spatial derivatives of the reverse field but also on time regularity, such as $\partial_t \nabla \log p_t(x)$. Moreover, since $r_k$ is evaluated along perturbed trajectories, it is also influenced by the score-estimation error through the shifted states. We therefore view a joint bound on total sampling error as a natural but technically nontrivial direction for future work.
>
> ## Real-data validation beyond Gaussian mixtures
>
> We add a new CIFAR-10 experiment in the revision to address the concern about real-data validation. A subset of CIFAR-10 is treated as an empirical distribution, whose Gaussian-noised marginal is a finite Gaussian mixture. Over the last $K$ reverse steps near the data end, we replace the exact empirical score by a learned score, while keeping the exact empirical score for the earlier steps, and evaluate the final discrepancy using sliced Wasserstein distance.
>
> The results show the same qualitative trend as in Figure 1: schedules with stronger data-end refinement are more robust to injected score error. In particular, uniform log-SNR performs best across all tested settings, while the linear schedule deteriorates the fastest as $K$ increases.
>
> | Cutoff $K$ | Linear | Quadratic | Cosine | Uniform log-SNR |
> |---:|---:|---:|---:|---:|
> | 100 | 0.2284 | 0.0617 | 0.0869 | **0.0401** |
> | 200 | 0.6858 | 0.1657 | 0.2329 | **0.0461** |
> | 300 | 1.4101 | 0.3866 | 0.4289 | **0.0912** |
> | 400 | 2.5693 | 0.7511 | 0.7105 | **0.3811** |
>
> We will include these results in the revision.

---

> > ### Author Rebuttal · Reviewer_6PjJ · 2026-04-02
> >
> > Thank you for the detailed rebuttal. The added CIFAR-10 experiment is helpful, although it remains somewhat controlled. I think some clarification would be useful: in this experiment, the learned score is used only in the final reverse steps, while the exact score is used earlier in the trajectory. Could you comment on how representative this setup is of standard diffusion sampling, where the learned score is used throughout? In particular, why was this setup chosen?
> >
> > Regarding Assumption 3.1 and Corollary 3.14, I understand that $L_t$ (the Lipschitz constant of the score approximation error) and $C_0$ (which controls the curvature of the true smoothed density) are different quantities. However, since $L_t$ depends on the approximation error relative to $\nabla \log p_t$, I would expect some relationship between $L_0$ and the regularity of $\nabla \log p_0$, which is reflected in $C_0$. Could the authors clarify whether such a link exists?
> >
> > Could the authors also comment on the dependence of $C_0$ on $||\nabla \ell_0(0)||$? Since this term depends on the choice of reference point, it is not immediately clear how intrinsic or practically meaningful it is. Is this mainly a technical artifact of the proof, or does it have a natural interpretation in the refined small-time Hessian bound?

---

> > > ### Author Response · Authors · 2026-04-07
> > >
> > > We thank the reviewer for the helpful follow-up questions.
> > >
> > > ### the  experiment setup
> > >
> > > The CIFAR-10 experiment is intentionally designed as a controlled yet practically relevant setting. In standard diffusion sampling, although the learned score is used along the whole trajectory, the estimation error is not uniform. Near the noise end, the distribution is closer to Gaussian, so score estimation tends to be more accurate there, whereas larger estimation errors are more likely near the data end. We therefore replace the exact score with the learned score over segments of varying length anchored at the data end, so as to better illustrate how the effect of score-estimation error accumulates in the more sensitive region under different schedules.
> > >
> > >
> > >
> > >
> > > ### the relation between $L_t$ and $C_0$
> > >
> > > Regarding the relation between $L_t$ and $C_0$, there is an indirect but not deterministic link. Corollary 3.14 controls the curvature of the true score field near the data end, i.e., a quantity of the form
> > > $$
> > > \sup_x \|\nabla_x^2 \log p_t(x)\|.
> > > $$
> > > By contrast, $L_t$ is the Lipschitz constant of the score approximation error
> > > $$
> > > e_t(x) = s_\theta(x,t) - \nabla_x \log p_t(x).
> > > $$
> > > If $e_t$ is differentiable in $x$, then
> > > $$
> > > L_t \le \sup_x \|\nabla_x e_t(x)\|
> > > = \sup_x \bigl\|\nabla_x s_\theta(x,t) - \nabla_x^2 \log p_t(x)\bigr\|.
> > > $$
> > > Thus, $C_0$ controls the true-score curvature term, but $L_t$ also depends on the spatial regularity of the estimator $s_\theta(\cdot,t)$. In this sense, $C_0$ informs but does not by itself determine $L_t$.
> > >
> > > That said, if one is willing to impose additional derivative-level control on the estimator, then a more explicit relationship can indeed be written down. For example, if
> > > $$
> > > \sup_x \|\nabla s_\theta(x,t)\| \le B_t,
> > > $$
> > > then, combined with the small-$t$ curvature control from Corollary 3.14, one obtains
> > > $$
> > > L_t \le B_t + C_0
> > > $$
> > > for small $t$. Even more directly, if one assumes a $C^1$-level approximation bound
> > > $$
> > > \sup_x \|\nabla s_\theta(x,t)-\nabla^2 \log p_t(x)\| \le \delta_t,
> > > $$
> > > then one directly gets
> > > $$
> > > L_t \le \delta_t.
> > > $$
> > >
> > > ### The role of the reference point in Corollary 3.14
> > >
> > >
> > > The dependence of $C_0$ on $\|\nabla \ell_0(0)\|$ mainly comes from the way the proof makes the constant explicit, rather than from any intrinsic role of the origin. In the argument, one needs to anchor the gradient at a reference point when passing from second-derivative control to first-derivative control, and we chose $0$ only for convenience. More generally, for any fixed reference point $x_{\mathrm{ref}}$, if $\nabla \ell_0$ is $L$-Lipschitz, then
> > > $$
> > > \|\nabla \ell_0(x)\|
> > > \le
> > > \|\nabla \ell_0(x_{\mathrm{ref}})\| + L\|x-x_{\mathrm{ref}}\|,
> > > $$
> > > and therefore
> > >
> > > $$
> > > \mathbb E_{x_0\sim p_0}|\nabla \ell_0(x_0)|^2
> > > \le 2|\nabla \ell_0(x_{\mathrm{ref}})|^2 + 2L^2 \mathbb E_{x_0\sim p_0}|x_0-x_{\mathrm{ref}}|^2.
> > > $$
> > >
> > > So the appearance of $\|\nabla \ell_0(0)\|$ reflects one convenient anchoring choice in the proof, rather than a special role of the origin. We will clarify this point in the revision.

---

### Official Review · Reviewer_kyo8 · 2026-03-11

**Soundness:** 2
**Presentation:** 3
**Significance:** 2
**Originality:** 3
**Overall Recommendation:** 4
**Confidence:** 4

**Summary:**

This paper studies how score estimation errors accumulate through the reverse dynamics of diffusion models. Using synchronous coupling and a pathwise averaged Hessian-based gain recursion, the authors derive W_2​ bounds that resolve the per-step contribution of score error. The analysis proceeds from Gaussian to Gaussian mixtures even with different weights and covariance to general smooth distributions for both VP/VE and Probability flow odes. Experiments on 1-dimensional Gaussian mixtures confirm the theoretical results.

**Compliance With Llm Reviewing Policy:**

Affirmed.

**Final Justification:**

Given that the authors have addressed my concerns, I am raising my rating.

**Key Questions For Authors:**

- Could you justify Part 1 of Assumption 3.1? It requires the neural network to be Lipschitz, whereas much of the recent literature does not seem to rely on this assumption. What difficulty prevents removing it? Also, the monotonicity assumption appears to be nonstandard.

- In Line 608, you assume $E[e_{\tau_i}] = 0$ and independence without justification.

- In Lines 630, you assume the existence of constant $C_0$ without explanation.

**Limitations:**

yes

**Strengths And Weaknesses:**

Strength:

This paper studies an interesting problem of accumulation of score estimation error in diffusion models covering almost all settings, and give sharp bound for Guassian case. The surrogate Hessian method for mixture Gaussian case is interesting.

Weakness:

- some theoretical weakness, see questions

- Experiments are limited to 1-D Gaussian mixtures with no real-data validation, Assumption 3.1's monotonicity is presented as formalizing empirical trends but no empirical verification is provided, and cited references do not establish this property.

---

> ### Author Rebuttal · Authors · 2026-03-31
>
> We thank the reviewer for the detailed comments and questions. Below we provide responses to each point.
> ## Experiment
> Our current empirical validation in the main paper is conducted on a 10-dimensional Gaussian mixture, where the theoretical quantities are explicit and the comparison between theory and experiment can be made cleanly. To further address the concern about real-data validation, we have added a new CIFAR-10 experiment in the revision.
>
> In this experiment, a subset of CIFAR-10 is treated as an empirical distribution. Under Gaussian noising, its perturbed marginal becomes a finite Gaussian mixture, where we can use the exact empirical score for all $t>0$. We then run a controlled experiment in which, over the last $K$ reverse steps near the data end, we replace the exact empirical score by a learned score, while keeping the exact empirical score for the remaining earlier steps. We evaluate the final discrepancy using sliced Wasserstein distance.
>
> The results show the same qualitative trend as in Figure 1: schedules with stronger refinement near the data end are consistently more robust to injected score error. In particular, uniform log-SNR performs best across all tested settings, while the linear schedule performs worst. The results are as follows:
>
> | Cutoff $K$ | Linear | Quadratic | Cosine | Uniform log-SNR |
> |---:|---:|---:|---:|---:|
> | 100 | 0.2284 | 0.0617 | 0.0869 | **0.0401** |
> | 200 | 0.6858 | 0.1657 | 0.2329 | **0.0461** |
> | 300 | 1.4101 | 0.3866 | 0.4289 | **0.0912** |
> | 400 | 2.5693 | 0.7511 | 0.7105 | **0.3811** |
>
> We will include these results in the revision to demonstrate that the mechanism identified by our theory is not limited to low-dimensional synthetic mixtures, but also appears on real image data.
>
> ## Assumption 3.1
>
> Assumption 3.1(i) is not an assumption that the neural network $s_\theta(x,t)$ itself must be Lipschitz. What we directly assume is that the score approximation error $e(x,t)=s_\theta(x,t)-\nabla\log p_t(x)$ is $L_t$-Lipschitz. The discussion below the assumption was only intended to give a sufficient condition in a structured setting: when $p_0$ is a Gaussian mixture, the ground-truth score $\nabla \log p_t$ is globally Lipschitz for every $t>0$, so if $s_\theta(\cdot,t)$ is Lipschitz, then $e(\cdot,t)$ is also Lipschitz. We will revise the text to make this distinction explicit. Second, the reason for Assumption 3.1(i) is technical. Our analysis needs to control the difference $e(y_k,\tau_k)-e(y_k^{(0)},\tau_k)$ along the coupled trajectories. A Lipschitz bound on $e(x,t)$ allows us to relate this term to $|y_k-y_k^{(0)}|$ and close the gain recursion.
>
> As for the monotonicity assumption, it is heuristic rather than standard. Its role is to encode the empirically observed trend that score approximation is typically more difficult near the data end and easier at higher noise levels. When $t$ is small, the data distribution is less smoothed and the approximation error is usually larger; when $t$ is large, the marginal $p_t$ is more heavily smoothed and closer to Gaussian, so the score field varies more gently and is easier to approximate. Motivated by this, we model both $L_t$ and $\varepsilon_t$ as non-increasing in $t$. We will revise the manuscript to make clear that this is a modeling assumption motivated by empirical behavior.
>
> ## Clarification on $E[e_{\tau_i}]$
>
> The conditions $\mathbb{E}[e_{\tau_i}]=0$ and independence across $i$ are not part of Assumption 3.1 and should not appear implicitly in the proof. More importantly, they are not needed for the bound. Indeed, for
> $S_1=\sum_{i=0}^{K-1} G_i(\bar H)e_{\tau_i}$,
> one can directly use
> $
> \lVert S_1\rVert \le \sum_{i=0}^{K-1} \lVert G_i(\bar H)\rVert_{\mathrm{op}} \, \lVert e_{\tau_i}\rVert
> $
> followed by the Cauchy--Schwarz inequality in the form
> $(\sum_i a_i)^2 \le K \sum_i a_i^2$,
> together with Assumption 3.1(ii), to obtain
> $\mathbb{E}\|S_1\|^2 \le K \sum_{i=0}^{K-1} g_i(\bar H)^2 \varepsilon_{\tau_i}^2$
> without any zero-mean or independence assumption. We will revise the proof accordingly and remove this unnecessary step.
>
> ## Clarification on the constant $C_0$
> In the proof of Theorem 3.7, the constant $C_0$ was only intended as a shorthand for a uniform one-step bound, rather than as an additional assumption. More concretely, in the Gaussian-mixture setting, the relevant Hessian terms are controlled by the component covariance inverses, and the error Jacobian is controlled by $L_{\tau_l}$ from Assumption 3.1.
> Hence
> $\operatorname{opnorm}(\alpha_l I_d + \beta_l(H_l + \overline{E})) \le \alpha_l + \beta_l(\Lambda + L{\tau_l}) \le 1 + \beta_l(\Lambda + L_{\tau_l}) =: C_0$
> We will revise the appendix to make this step explicit and remove the current ambiguous wording suggesting that $C_0$ is an extra assumption.

---

> > ### Author Rebuttal · Reviewer_kyo8 · 2026-04-04
> >
> > Thank you for the detailed response. I will raise my rating.

---

> > > ### Author Response · Authors · 2026-04-07
> > >
> > > Thank you very much for your careful reading and thoughtful feedback. We are glad that the concerns you raised have been addressed. We sincerely appreciate your time and valuable comments.

---

### Official Review · Reviewer_MVwC · 2026-03-13

**Soundness:** 3
**Presentation:** 3
**Significance:** 3
**Originality:** 4
**Overall Recommendation:** 5
**Confidence:** 3

**Summary:**

Paper analyzes score estimation error accumulation in diffusion models during sampling. It establishes non-asymptotic Wasserstein distance bounds for variance-preserving and variance-exploding SDEs and probability-flow ODEs using first-order methods. Analysis derives exact pathwise Hessians for Gaussian distributions (Theorem 3.5), surrogate Hessians for Gaussian mixtures (Theorem 3.7), and operator-norm curvature envelopes for general distributions (Theorem 3.12). Results prove discretization errors amplify heavily near the data end (t = 0), mathematically justifying the empirical success of cosine and uniform log-SNR noise schedules.

**Compliance With Llm Reviewing Policy:**

Affirmed.

**Final Justification:**

I recommend acceptance.

My initial high score (5) was further reinforced by the rebuttal.

**Key Questions For Authors:**

1. How do the error bounds in Theorem 3.7 and 3.12 evolve when applied to second-order or higher-order ODE solvers common in modern pipelines? If the authors can demonstrate that their bounds easily extend to higher-order solvers without blowing up, this would raise Significance score.
2. Corollary 3.14 assumes a globally Lipschitz score. What empirical or theoretical evidence justifies this assumption for complex, non-Gaussian diffusion models? If the authors can provide empirical evidence or a theoretical relaxation showing this assumption holds (or isn't strictly necessary) for datasets like CIFAR-10, it would improve the Soundness score.
3. Can the theoretical bounds be evaluated on standard image datasets (e.g., CIFAR-10) to demonstrate their practical tightness and scalability? Providing even a small-scale empirical validation on real image data would help.

**Limitations:**

yes

**Strengths And Weaknesses:**

Soundness:

Strength: Mathematical derivations for Gaussian and Gaussian mixture distributions are rigorous. Isolating multiplicative amplification governed by curvature matrices $H_j = -\Sigma_{\tau_j}^{-1}$ and additive injection from local score errors (Equation 11, Theorem 3.5) provides good control.
Weakness: Theorem 3.12 relies on a curvature envelope that blows up as $\tau \to 0$. Mitigating this via Corollary 3.14 requires assuming a globally Lipschitz score, which is unrealistic for complex real-world data manifolds.
Weakness: Empirical validation is restricted to a 10-dimensional synthetic symmetric Gaussian mixture (Figure 1). This is insufficient to demonstrate theoretical tightness on large-scale, high-dimensional datasets.

Presentation:
Strength: Logical progression from exact pathwise Hessians in simple Gaussian cases to surrogate Hessians and general bounding expectations is clear. Distinction between small and large separation regimes is explicit.
Weakness: Reliance on synthetic datasets in Figure 1 obscures how these bounds practically translate to standard image generation tasks.

Significance:

Strength: Grounding the empirical success of data-end-heavy step sizes (e.g., cosine scheduling) in formal mathematical proofs rather than heuristics is relevant. It provides a principled basis for sampler design.
Weakness: Scope is strictly limited to first-order samplers. Modern applications predominantly use higher-order ODE solvers. Findings fail to offer new scheduling algorithms beyond confirming existing intuition.

Originality:

Strength: Utilizing pathwise Hessian averages to bound score error propagation deviates significantly from conventional uniform L2 error bounds. Establishing regime-adapted surrogate Hessians (average and dominant-component) for Gaussian mixtures is highly creative and a distinct theoretical advancement.

---

> ### Author Rebuttal · Authors · 2026-03-31
>
> We thank the reviewer for thoughtful comments and questions. Below we provide responses to each point.
>
> ### Corollary 3.14 / blow-up
>
> Corollary 3.14 is not intended as a literal model for complex real-world data manifolds, but rather as one sufficient condition under which the small-time blow-up in the distribution-free baseline of Theorem 3.12 can be improved. By contrast, Gaussian mixture models provide a useful structured approximation for certain real-world data manifolds, provided the approximation uses non-degenerate covariance components. From this perspective, our Gaussian-mixture analysis offers a complementary surrogate view beyond the fully distribution-free setting. In particular, in the GMM case, error accumulation is described explicitly through the component covariance inverses $(\Sigma_k^{-1})$, rather than only through the coarse worst-case smoothing-scale envelope in Theorem 3.12.
>
> ---
>
> ### Image dataset validation
>
>  We have added a new CIFAR-10 experiment in the revision. In this experiment, a subset of CIFAR-10 is treated as an empirical distribution. Under Gaussian noising, its perturbed marginal becomes a finite Gaussian mixture, where we can use the exact empirical score for all $t>0$. We then run a controlled experiment in which, over the last $K$ reverse steps near the data end, we replace the exact empirical score by a learned score, while keeping the exact empirical score for the remaining earlier steps. We evaluate the final discrepancy using sliced Wasserstein distance.
>
> The results show the same qualitative trend as in Figure 1: schedules with stronger refinement near the data end are consistently more robust to injected score error. In particular, uniform log-SNR performs best across all tested settings, while the linear schedule performs worst. The results are as follows:
>
> | Cutoff $K$ | Linear | Quadratic | Cosine | Uniform log-SNR |
> |---:|---:|---:|---:|---:|
> | 100 | 0.2284 | 0.0617 | 0.0869 | **0.0401** |
> | 200 | 0.6858 | 0.1657 | 0.2329 | **0.0461** |
> | 300 | 1.4101 | 0.3866 | 0.4289 | **0.0912** |
> | 400 | 2.5693 | 0.7511 | 0.7105 | **0.3811** |
>
> We will include these results in the revision to demonstrate that the mechanism identified by our theory is not limited to low-dimensional synthetic mixtures, but also appears on real image data.
>
> ---
>
> ### Higher-order samplers
>
> Our current scope is limited to first-order samplers because the analysis only requires Hessian-level control. A higher-order sampler extension would need bounds on higher-order score derivatives, e.g. $\mathbb{E}\|\nabla^3 \log p_t(X_t)\|_{\mathrm{op}}$, which are much harder to obtain. In fact, without additional assumptions, such quantities are expected to be even more singular near the data end than the Hessian bound in Lemma B.1. For this reason, we do not claim a higher-order result under the present assumptions. However, if higher-order score derivatives can be controlled sharply enough, our propagation framework could still be applicable to higher-order samplers.
>
> ---
>
> ### Lipschitz-score assumption
>
> The role of Corollary 3.14 is only to show that the small-time blow-up in the distribution-free baseline of Theorem 3.12 can improve under additional regularity.
>
> At the same time, this type of regularity does hold in the structured settings analyzed in the paper. In particular, for Gaussian and Gaussian-mixture marginals with non-degenerate covariance components, the score is globally Lipschitz for every $t>0$; see Lemma B.3. This is also the regime relevant to our CIFAR-10 experiment with early stopping. If a finite CIFAR-10 subset is treated as an empirical distribution, then for every fixed $\tau>0$, the noisy marginal on $[\tau,T]$ is a finite Gaussian mixture with non-degenerate covariance components. Thus, while we do not claim that the original data distribution at $t=0$ satisfies Corollary 3.14, the regularized marginals beyond the early stopping time $\tau$ do fall into a structured regime where the score is well behaved. In this sense, early stopping avoids the most singular data-end regime while still allowing us to test the qualitative predictions of the theory on real images.

---

> > ### Author Rebuttal · Reviewer_MVwC · 2026-04-03
> >
> > Thank you.
> >
> > The rebuttal adequately addresses my concerns. I appreciate that the authors added a real-image CIFAR-10 experiment, which strengthens the empirics support beyond GMMs and shows the same qualitative schedule ranking predicted by the theory. The clarification around Corollary 3.14 is also helpful. The response on higher-order samplers is reasonable. My remaining reservation is therefore mainly about scope, not correctness.
> >
> > My initial scores were already high, so I will not increase them further.

---

> > > ### Author Response · Authors · 2026-04-07
> > >
> > > Thank you very much for your careful reading and thoughtful feedback. We are glad that our response has addressed your concerns. We sincerely appreciate your time and valuable comments.

---

### Decision · Program_Chairs · 2026-04-30

**Decision:**

Accept (regular)

**Comment:**

The reviewers agreed that this submission makes a technically solid and original theoretical contribution by giving a per-step analysis of how score-estimation errors accumulate during diffusion sampling, with a clear progression from Gaussian to Gaussian-mixture to more general settings, and by formalizing why schedules that allocate more refinement near the data end are more robust. At the same time, several reviewers noted limitations in scope and practical impact: the strongest results are obtained in structured Gaussian/Gaussian-mixture regimes, the general-distribution bound appears conservative and may rely on regularity conditions that are difficult to justify broadly, the analysis is restricted to first-order samplers, and the initial empirical validation was limited. The rebuttal addressed a substantial portion of these concerns by clarifying the role of the assumptions, correcting proof-level ambiguities, and adding a controlled CIFAR-10 experiment that supports the main qualitative claim about schedule robustness; after rebuttal. Overall, the paper offers a meaningful theoretical advance with clear exposition, though it would be valuable to understand how the analysis might extend to less benign settings in which Assumption 3.1 does not hold; one important example is the data manifold regime, where the ground-truth score becomes singular in the small-noise limit.